# Heat shock factor regulation of antimicrobial peptides expression suggests a conserved defense mechanism induced by febrile temperature in arthropods

Bang Xiao[1,2], Shihan Chen[1,2], Yue Wang[1,2], Xuzheng Liao[1,2], Jianguo He[1,2,3]*, Chaozheng Li[1,2,3]*

[1]School of Marine Sciences, Sun Yat-sen University, State Key Laboratory of Biocontrol /Southern Marine Science and Engineering Guangdong Laboratory (Zhuhai), Guangzhou, China; [2]Guangdong Provincial Key Laboratory of Marine Resources and Coastal Engineering/ Guangdong Provincial Key Laboratory for Aquatic Economic Animals, School of Life Sciences, Sun Yat-sen University, Guangzhou, China; [3]China-ASEAN Belt and Road Joint Laboratory on Mariculture Technology, Guangzhou, China

**\*For correspondence:**
lsshjg@mail.sysu.edu.cn (JH);
lichaozh@mail2.sysu.edu.cn (CL)

**Competing interest:** The authors declare that no competing interests exist.

## eLife Assessment

This is an **important** study that addresses the role of fever as a conserved response to viral infection. It demonstrates that the heat-shock factor, HSF1, is activated by increased temperature during fever to enhance the anti-viral immune response. The data provides **compelling** evidence for the conclusions and the work will be of interest to virologists, immunologists, and cell biologists.

**Abstract** Temperature is a critical factor influencing the outbreak and progression of viral diseases in organisms. Febrile temperatures have been shown to enhance immune competence and reduce viral replication in various species. However, the underlying mechanisms remain largely unknown. In this study, we investigate the molecular mechanisms by which elevated temperatures confer resistance to viral infections, focusing on the role of heat shock factor 1 (HSF1) in regulating antimicrobial effectors rather than the traditional target genes molecular chaperones. Using shrimp *Litopenaeus vannamei* as a model, we demonstrate that febrile temperatures induce HSF1, which in turn upregulates antimicrobial peptides (AMPs) that target viral envelope proteins and inhibit viral replication. Importantly, this is the first to show that HSF1 directly binds to the heat shock element (HSE) motifs of AMPs both in shrimp and *Drosophila* S2 cells, suggesting this may be a conserved regulatory mechanism in arthropods. Additionally, our findings highlight the role of HSF1 beyond the classical heat shock response, revealing its critical function in modulating innate immunity. These insights provide new avenues for managing viral infections in aquaculture and other settings by leveraging environmental temperature control.

## Introduction

The proliferation of pathogens and the outbreak of diseases are influenced by various factors, including transmission vectors, host species, and environmental conditions (*Marcogliese, 2008*). Among these, temperature is one of the most critical environmental factors, significantly impacting disease occurrence and prevalence (*Hou et al., 2023*). Temperature can directly affect the immune response, metabolism, feeding rate, oxygen consumption, growth, and survival of animals (*Deldicq et al., 2021*). Aquatic animals such as fish, shrimp, crabs, and shellfish are ectothermic, and their susceptibility to diseases is directly related to temperature (*Marcogliese, 2008*). In fishes, Koi herpesvirus (KHV) can cause massive mortality in carp and koi carp at temperatures between 23°C and 28°C, while temperatures above 30 °C typically do not result in mortality (*Yuasa et al., 2008*). In crustaceans, high temperatures (32–33°C) have been demonstrated to inhibit white spot syndrome virus (WSSV) replication and reduce mortality in infected shrimp (*Du et al., 2006*; *Guan et al., 2003*; *Jiravanichpaisal et al., 2004*). Notably, many ectothermic animals can respond to exogenous pathogenic infections by seeking warmer external environments (*Covert and Reynolds, 1977*; *Louis et al., 1986*). When infected, some ectothermic animals such as fish, insects, and crustaceans elevate their body temperature by moving to warmer areas (*Bundey et al., 2003*; *Rakshaninejad et al., 2023*; *Rakus et al., 2017*). This behavioral fever, similar to fever responses in higher organisms, can help hosts suppress or overcome infections (*Hasday et al., 2014*), although the mechanisms underlying this heat-induced antimicrobial immunity remain unclear.

The heat shock response (HSR), also known as the heat stress response, serves as a bridge linking elevated temperatures with increased immunity (*Tian et al., 2021*). The HSR induced by temperature increase primarily involves the expression of heat shock proteins (HSPs) and other related factors regulated by heat shock transcription factors (HSFs; *Lang et al., 2021*). Under non-stress conditions, HSF monomers are retained in the cytoplasm in a complex with HSP90 or the cytosolic chaperonin TCP1 ring complex (TRiC; *Neef et al., 2014*). During the stress response, HSF dissociates from the complex, trimerizes, and converts into a DNA-binding conformation (*Zheng et al., 2016*). Activated HSF induces gene transcription through regulatory upstream promoter elements known as heat shock elements (HSEs; *Andrási et al., 2021*). In the face of thermal or pathogenic stress, the HSR functions as a highly conserved cellular mechanism across various organisms (*Anckar and Sistonen, 2011*). Research suggests that the HSF-mediated HSR plays a crucial role in the regulation of the innate immune network through a combination of direct gene regulation, interaction with immune signaling pathways, and the broader cellular stress response (*Morimoto, 2011*). In *Drosophila*, HSF-deficient flies, lacking a HSR, exhibit high sensitivity to viral infection (*Merkling et al., 2015*), indicating that the HSR is a crucial component of innate antiviral immunity.

Heat shock factors (HSFs) are primarily recognized as inducible transcriptional regulators of genes encoding molecular chaperones and other stress proteins (*Akerfelt et al., 2010*). Interestingly, members of the HSF family also participate in various cellular processes, including apoptosis, the unfolded protein response (UPR), oxidative stress, multidrug resistance, and pathogenic infection (*Anckar and Sistonen, 2011*). Consequently, the repertoire of HSF targets has expanded well beyond traditional heat shock genes (*Barna et al., 2018*). Recent research has illuminated the interplay between the HSR and the innate immune system, revealing that HSFs can regulate innate immune responses (*Akerfelt et al., 2007*). For instance, HSF1, the primary HSF, can modulate the expression of pro-inflammatory cytokines such as TNF-α, IL-1β, and IL-6, thereby influencing the inflammatory response (*Barna et al., 2018*). Additionally, HSF1 has been shown to interact with and influence the activity of the NF-$\kappa$B and MAPK pathways, thus altering the cellular environment to modulate the innate immune response (*Chen et al., 2010*). These unexpected observations indicate the complexity of HSF function. At present, however, the mechanisms by which the HSF family affects innate immunity are still poorly understood.

To explore the underlying mechanisms of the febrile temperature-inducible HSF in host antiviral immunity, the shrimp *Litopenaeus vannamei* was challenged with WSSV, and the thermal regulation of host immunity was characterized. The results indicated that febrile temperature-inducible HSF1 confers virus resistance by regulating the expression of antimicrobial peptides (AMPs) in *L. vannamei*. Additionally, it was revealed that high temperature (30°C) restricts the replication of *Drosophila* C virus (DCV) in an invertebrate model system (*Drosophila* S2 cells) also mediated by the HSF1-AMPs

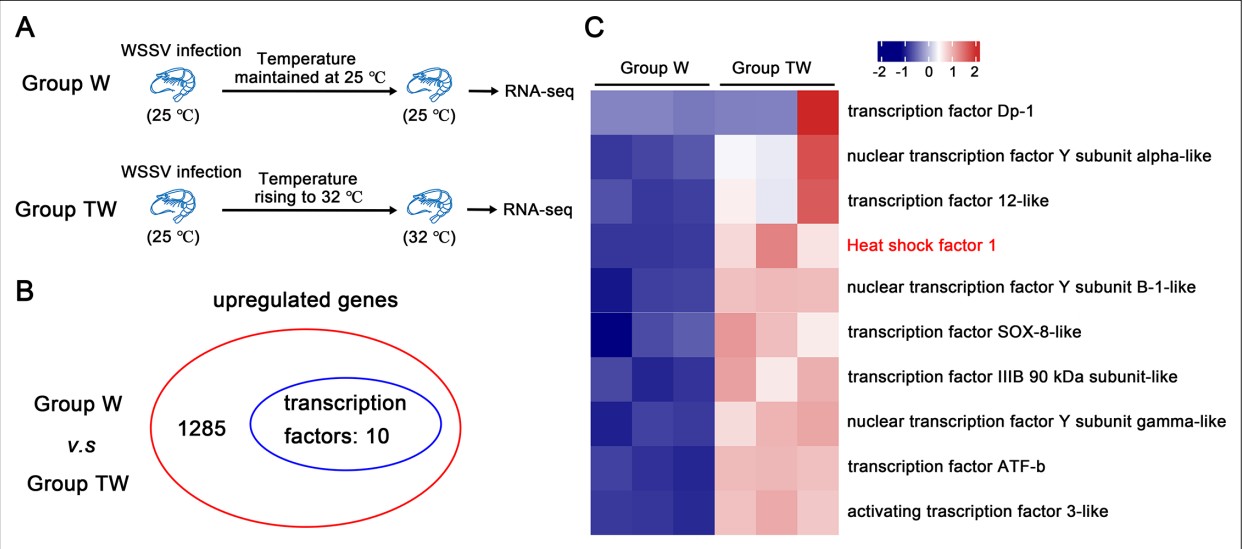

**Figure 1.** Identification of HSF1 as a key factor induced by high temperature. (**A**) Diagram illustration of the experimental setup for transcriptome analysis. Group W was injected with WSSV and maintained continuously at 25 °C. In contrast, Group TW, also injected with WSSV, had their culture temperature increased to 32 °C at 24 hours post-injection (hpi). (**B**) Identification of high temperature-induced genes by RNA-Seq. RNA sequencing was performed 12 hr post-temperature increase. Gill samples from eight shrimp in each group were collected for Illumina sequencing (NCBI SRA database accession number PRJNA1050424). (**C**) Heatmap showing the differential expression of upregulated genes encoding potential transcription factors. The heatmap displays the fragments per kilobase per transcript per million mapped reads (FPKM) values from three biological replicates, shown after logarithmic transformation.

The online version of this article includes the following source data and figure supplement(s) for figure 1:

**Source data 1.** Numerical source data for heatmap shown in *Figure 1C*.

**Figure supplement 1.** The expression of transcription factors as determined by qPCR.

**Figure supplement 1—source data 1.** Numerical source data for graphs shown in *Figure 1—figure supplement 1*.

axis. Together, these results suggest that HSF regulation of AMPs expression is a conserved defense mechanism induced by febrile temperature in arthropods.

## Results

### Induction of HSF1 at high temperature (32°C)

Studies have demonstrated that high temperatures (above 30 °C) markedly reduce mortality in shrimp infected with WSSV, while temperatures ranging from 22°C to 30 °C favor higher levels of WSSV replication and associated mortality (*Lin et al., 2011*; *Rahman et al., 2007*; *Xiao et al., 2024*). To uncover the molecular mechanisms by which high temperature restricts WSSV infection, RNA-seq was performed to identify genes responsive to high temperature, particularly those encoding potential transcriptional regulators. Two shrimp groups, Group TW and Group W, were cultured at 25 °C. Group W comprised shrimp injected with WSSV and maintained at 25 °C continuously. In contrast, Group TW was subjected to a temperature increase to 32 °C at 24 hours post-injection (hpi). Gill samples were collected for analysis 12 hours post-temperature rise (hptr) and subjected to Illumina sequencing (*Figure 1A*). As shown in *Figure 1B and C*, 10 of the 1285 high temperature-induced genes were predicted to encode transcription factors.

GO functional enrichment analysis of DEGs between group TW and group W, indicating that most DEGs were involved in biological processes such as protein refolding, chaperone-mediated protein folding, and heat response (*Xiao et al., 2024*). Special attention has been paid to HSF1, the master regulator of the HSR, which upon activation induces not only classical heat shock genes but also a broad array of genes beyond the scope of molecular chaperones (*Barna et al., 2018*). To validate the high-throughput sequencing results, the expression profile of these transcription factors was confirmed by qPCR, showing a significant increase in *Litopenaeus vannamei* HSF1 (LvHSF1)

expression (*Figure 1—figure supplement 1*). These results suggest that LvHSF1 may play a potential role in enhancing shrimp resistance to WSSV at elevated temperatures (32 °C).

## Expression profiles of LvHSF1 in shrimp under varied temperature conditions and WSSV challenge

To determine that LvHSF1 is related to high temperature and WSSV infection, its expression level was measured under high temperature and after WSSV infection. Tissue distribution analysis showed that LvHSF1 was expressed in all examined tissues at room temperature (25 °C) and increased rapidly in all tested tissues at high temperature (32 °C; *Figure 2A*). Additionally, the protein expression of LvHSF1 was significantly up-regulated in hemocytes, gills, and intestines when exposed to 32 °C (*Figure 2B*). Next, we detected the expression of LvHSF1 in the immune response to Poly (I:C) and WSSV at 25 °C. Following exposure to Poly (I:C) at 25 °C, LvHSF1 expression was upregulated at 12 hr (2.43-fold), 24 hr (2.60-fold), and 36 hr (4.33-fold) in hemocytes (*Figure 2—figure supplement 1A*); slightly upregulated at 4 hr (1.97-fold) and downregulated at 12 hr (0.64-fold) in gills (*Figure 2—figure supplement 1B*); and slightly downregulated at 4 hr (0.41-fold) and upregulated at 8 hr (1.15-fold) in intestines (*Figure 2—figure supplement 1C*). In response to WSSV infection at 25 °C, LvHSF1 expression in hemocytes was dramatically induced, peaking at 8 hr (6.04-fold; *Figure 2—figure supplement 1D*). In the gills, LvHSF1 expression initially increased at 8 hr (2.10-fold) and significantly again at 72 hr (2.80-fold; *Figure 2—figure supplement 1E*). In the intestines, a marked induction of LvHSF1 was observed at 8 hr (2.23-fold; *Figure 2—figure supplement 1F*).

Further investigation revealed LvHSF1's role in the immune response to Poly (I:C) and WSSV at 32 °C. Following exposure to Poly (I:C) at 32 °C, LvHSF1 expression was upregulated at 4 hr (5.68-fold), 12 hr (8.44-fold), and 24 hr (7.52-fold) in hemocytes (*Figure 2C*); strongly induced at 24 hr (7.00-fold) in gills (*Figure 2D*); and slightly upregulated at 8 hr (3.47-fold) and 14 hr (2.60-fold) in intestines (*Figure 2E*). In response to WSSV infection at 32 °C, LvHSF1 expression in hemocytes was dramatically induced, peaking at 36 hr (15.93-fold; *Figure 2F*). In the gills, LvHSF1 expression initially increased at 8 hr (3.9-fold) and significantly again at 48 hr (4.56-fold; *Figure 2G*). In the intestines, the expression level of HSF1 remained at a high level from 8 to 48 hr (3.20~4.17 fold; *Figure 2H*). These results suggest that LvHSF1 is actively involved in the response to low (25 °C) and high (32 °C) temperature, and viral (WSSV) infection.

## Role of LvHSF1 in restricting WSSV at low (25 °C) and high (32 °C) temperatures

To investigate LvHSF1's role in shrimp during WSSV infection, RNA interference (RNAi) was employed in vivo. Two double-stranded LvHSF1 (dsLvHSF1) constructs were synthesized: dsLvHSF1-1 (targeting 443–1001 bp) and dsLvHSF1-2 (targeting 1365–1854 bp). Analysis revealed a significant reduction in both transcription and protein levels of LvHSF1 in hemocytes (*Figure 3A and a*) and gills (*Figure 3B and b*), with dsLvHSF1-1 showing greater silencing efficiency. Consequently, dsLvHSF1-1 was selected for subsequent knockdown experiments. After LvHSF1 knockdown, shrimp were challenged with WSSV, and viral titers were quantified using absolute quantitative PCR. Shrimp with silenced LvHSF1 exhibited increased viral loads in hemocytes (*Figure 3C*) and gills (*Figure 3D*) at low temperature (25 °C). Additionally, transcription levels of VP28, one of the major structural proteins of WSSV, were higher in the LvHSF1-silenced group than in the GFP-silenced control in hemocytes (*Figure 3E*) and gills (*Figure 3F*). Similarly, the levels of VP28 protein were significantly elevated in the HSF1-silenced group (*Figure 3G and g*). However, no significant difference in shrimp survival rates was observed between LvHSF1-silenced shrimp and GFP-silenced shrimp at low temperature (25 °C; p=0.8657; *Figure 3H*).

Next, LvHSF1's role in shrimp during WSSV infection at high temperature (32 °C) was further determined. Notably, the survival rate of LvHSF1-silenced shrimp infected with WSSV was significantly lower than the control group at high temperature (32 °C; p<0.001; *Figure 4A*). Shrimp with silenced LvHSF1 exhibited increased viral loads in gills (*Figure 4B*) and hemocytes (*Figure 4C*) at high temperature (32 °C). Additionally, VP28 protein levels were significantly elevated in the LvHSF1-silenced group in gills at high temperature (32 °C; *Figure 4D and d*). These data suggest that LvHSF1 plays a crucial role in antiviral immunity against WSSV infection, especially at high (32 °C) temperatures.

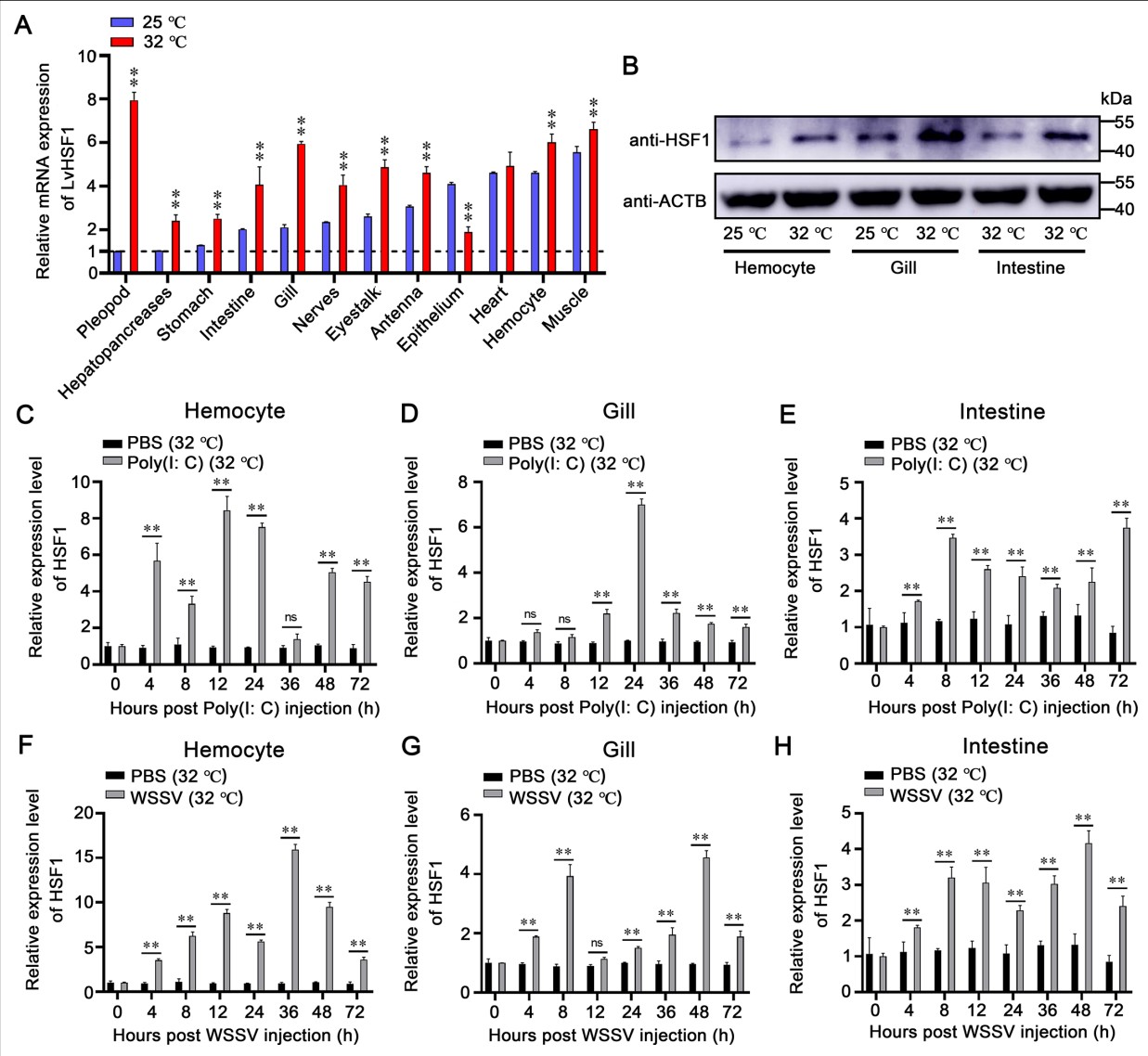

**Figure 2.** Upregulation of LvHSF1 in shrimp challenged by WSSV at both low (25 °C) and high (32 °C) temperatures. (**A**) Transcriptional expression of LvHSF1 in different tissues of healthy shrimp subjected to low (25 °C) and high (32 °C) temperatures for 12 hr. The pleopod tissue at 25 °C serves as a reference, with its expression level set to 1.0. Expression values were normalized against EF-1α and β-Actin using the Livak ($2^{-\Delta\Delta CT}$) method. Data represent the mean ± SD from triplicate assays. (**B**) Protein levels of HSF1 at 25 °C and 32 °C detected by western blotting. (**C–E**) Expression profiles of LvHSF1 after Poly (**I:C**) injection at high temperatures (32 °C) in hemocytes (**C**), gills (**D**), and intestines (**E**). Expression levels were measured by qPCR, normalized against EF-1α and β-Actin using the Livak ($2^{-\Delta\Delta CT}$) method, and presented as mean ± SD from triplicate assays. (**F–H**) Expression profiles of LvHSF1 after WSSV injection at high temperatures (32 °C) in hemocytes (**F**), gills (**G**), and intestines (**H**). Expression levels were measured by qPCR, normalized against EF-1α and β-Actin using the Livak ($2^{-\Delta\Delta CT}$) method, and presented as mean ± SD from triplicate assays. Statistical significance was calculated using the Student's $t$-test (**$p<0.01$, *$p<0.05$). All experiments were representative of three biological replicates and yielded similar results.

The online version of this article includes the following source data and figure supplement(s) for figure 2:

**Source data 1.** Numerical source data for graphs shown in *Figure 2A, C, D, E, F, G and H*.

**Source data 2.** TIF file with original western blots and boxes indicating the relevant bands shown in *Figure 2B*.

**Source data 3.** Original files for western blot analysis displayed in *Figure 2B*.

**Figure supplement 1.** The expression of profiles of LvHSF1 after Poly (I:C) injection at low temperature (25 °C).

**Figure supplement 1—source data 1.** Numerical source data for graphs shown in *Figure 2—figure supplement 1A, B, C, D, E and F*.

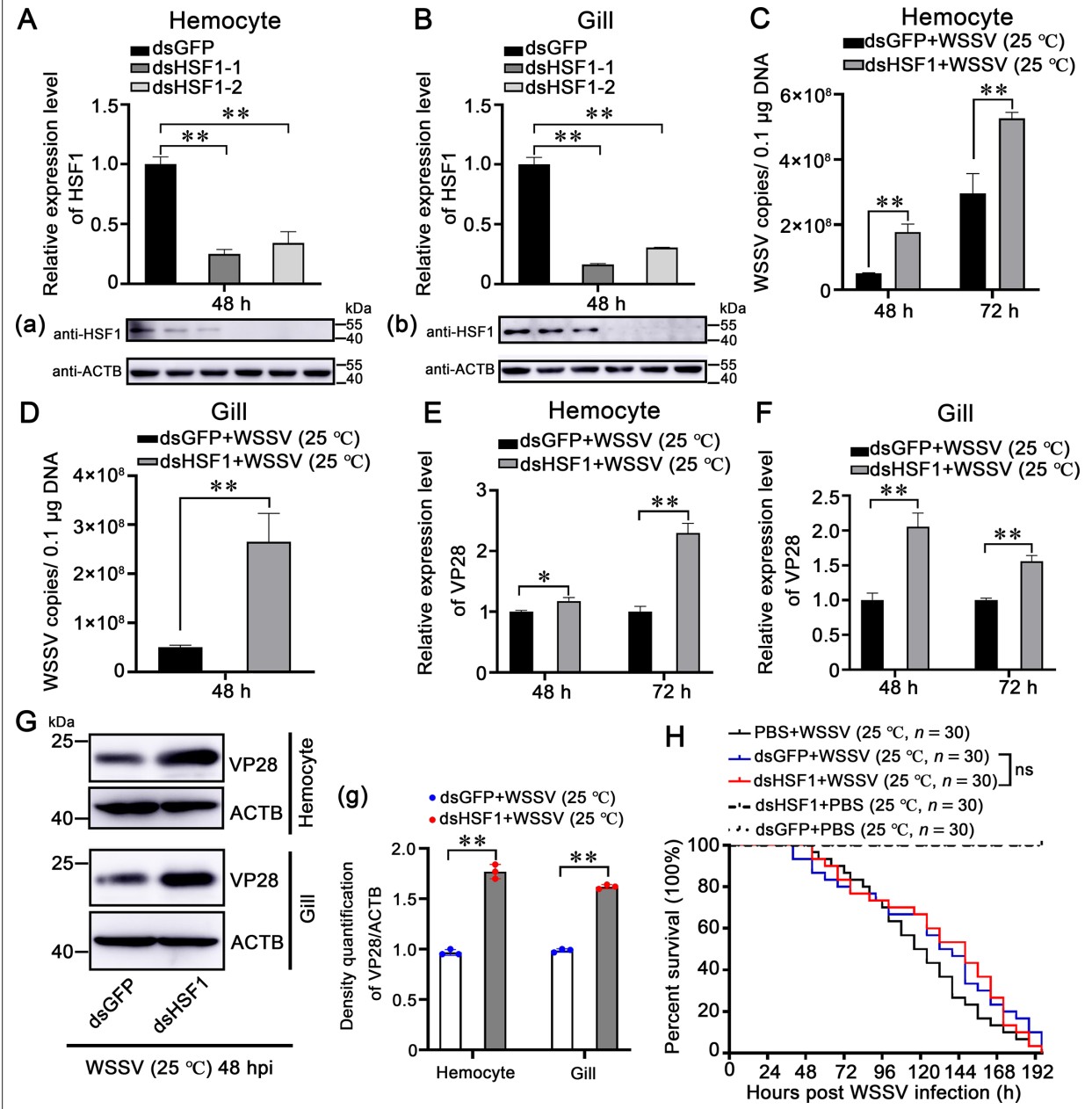

**Figure 3.** Role of HSF1 in restricting WSSV replication in shrimp at low (25 °C) temperatures. (**A, B**) Silencing Efficiency of dsRNA-LvHSF1. The efficacy of two dsRNA-LvHSF1 constructs (dsLvHSF1-1 and dsLvHSF1-2) was assessed by qPCR in hemocytes (**A**) and gills (**B**). Additionally, protein levels of LvHSF1 were evaluated by western blotting in both tissues (panels a and b). Expression values were normalized against EF-1α and β-Actin using the Livak (2$^{-\Delta\Delta CT}$) method. Data represent the mean ± SD from triplicate assays. (**C, D**) Quantification of WSSV copies. The quantity of WSSV copies in hemocytes (**C**) and gills (**D**) at low temperature (25 °C) post-WSSV infection was determined using absolute quantitative PCR. (**E, F**) Transcriptional expression of the VP28 gene. The transcriptional expression of the VP28 gene was analyzed by qPCR in hemocytes (**E**) and gills (**F**) at low temperature (25 °C) following WSSV infection. (**G**) Expression of WSSV VP28 protein. Expression of WSSV VP28 protein was detected by western blotting in hemocytes and gills at low temperature (25 °C) after WSSV infection. Quantification and statistical analysis of three independent repeats were performed using ImageJ (panel g). (**H**) Survival rates of WSSV-infected shrimp post-LvHSF1 knockdown. The survival rates of WSSV-infected shrimp post-LvHSF1 knockdown were monitored at low temperature (25 °C), with recordings made every 4 hr. Statistical analysis was performed using the Kaplan-Meier plot (log-rank $\chi^2$ test). All experiments were conducted with three biological replicates, consistently yielding similar results.

The online version of this article includes the following source data for figure 3:

**Source data 1.** Numerical source data for graphs shown in *Figure 3A, B, C, D, E, F, g and H*.

**Source data 2.** TIF file with original western blots and boxes indicating the relevant bands shown in *Figure 3a, b and g*.

**Source data 3.** Original files for western blot analysis displayed in *Figure 3a, b and g*.

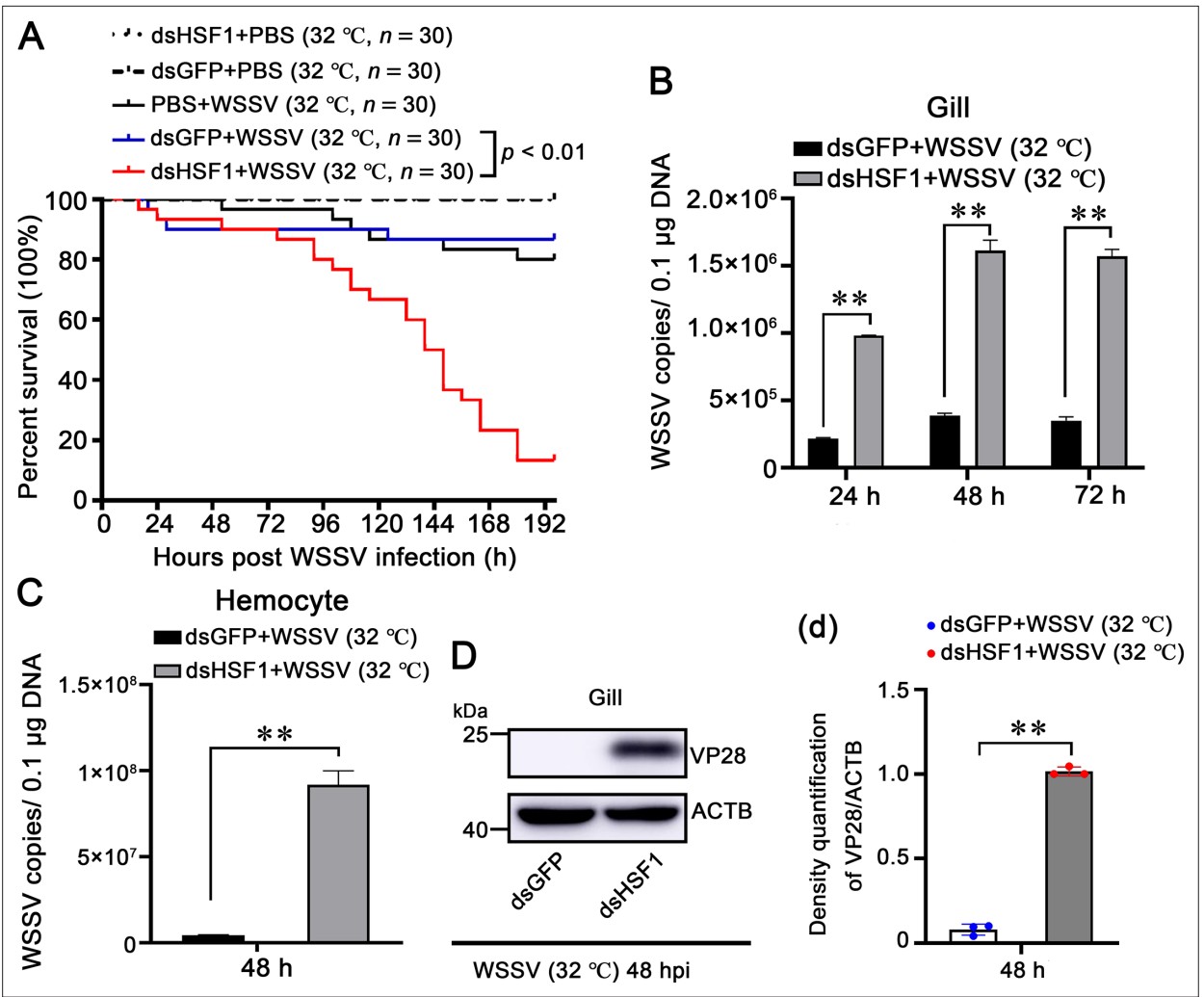

**Figure 4.** Role of HSF1 in restricting WSSV replication in shrimp at high (32°C) temperatures. (**A**) Survival rates of WSSV-infected shrimp post-LvHSF1 knockdown. The survival rates of WSSV-infected shrimp post-LvHSF1 knockdown were monitored at high temperature (32 °C), with recordings made every 4 hr. Statistical analysis was performed using the Kaplan-Meier plot (log-rank $\chi^2$ test). (**B, C**) Quantification of WSSV copies. The quantity of WSSV copies in gills (**B**) and hemocytes (**C**) at high temperature (32 °C) post-WSSV infection was determined using absolute quantitative PCR. (**D**) Expression of WSSV VP28 protein. Expression of WSSV VP28 protein was detected by western blotting in gills at high temperature (32 °C) after WSSV infection. Quantification and statistical analysis of three independent repeats were performed using ImageJ (panel d). Statistical significance was calculated using the Student's *t*-test (**p<0.01). All experiments were conducted with three biological replicates, consistently yielding similar results.

The online version of this article includes the following source data for figure 4:

**Source data 1.** Numerical source data for graphs shown in *Figure 4A, B, C and d*.

**Source data 2.** TIF file with original western blots and boxes indicating the relevant bands shown in *Figure 4D*.

**Source data 3.** Original files for western blot analysis displayed in *Figure 4D*.

## SWD as a potential antiviral effector regulated by LvHSF1

To determine how temperature-induced LvHSF1 restricts WSSV infection, RNA-seq was performed to identify target genes regulated by LvHSF1. Candidates whose expression was inhibited by LvHSF1 knockdown after WSSV infection at high temperature (32 °C) were screened. As shown in *Figure 5A and B*, 8 of the 954 dsHSF1-suppressed genes were predicted to encode effector factors. These differentially expressed genes (DEGs) included immunity-effector molecules, such as HSPs, AMPs, and cytokines, and the classical HSPs were downregulated by the RNA-seq (*Supplementary file 2*). Among the eight candidate genes, the anti-WSSV role of C-type lectins (CTLs), Anti-lipopolysaccharide factor (ALF), and Vago has been proved in shrimp (*Gao et al., 2021*; *Methatham et al., 2017*; *Zhao et al., 2009*). Therefore, particular attention was paid to the **n**ewly identified

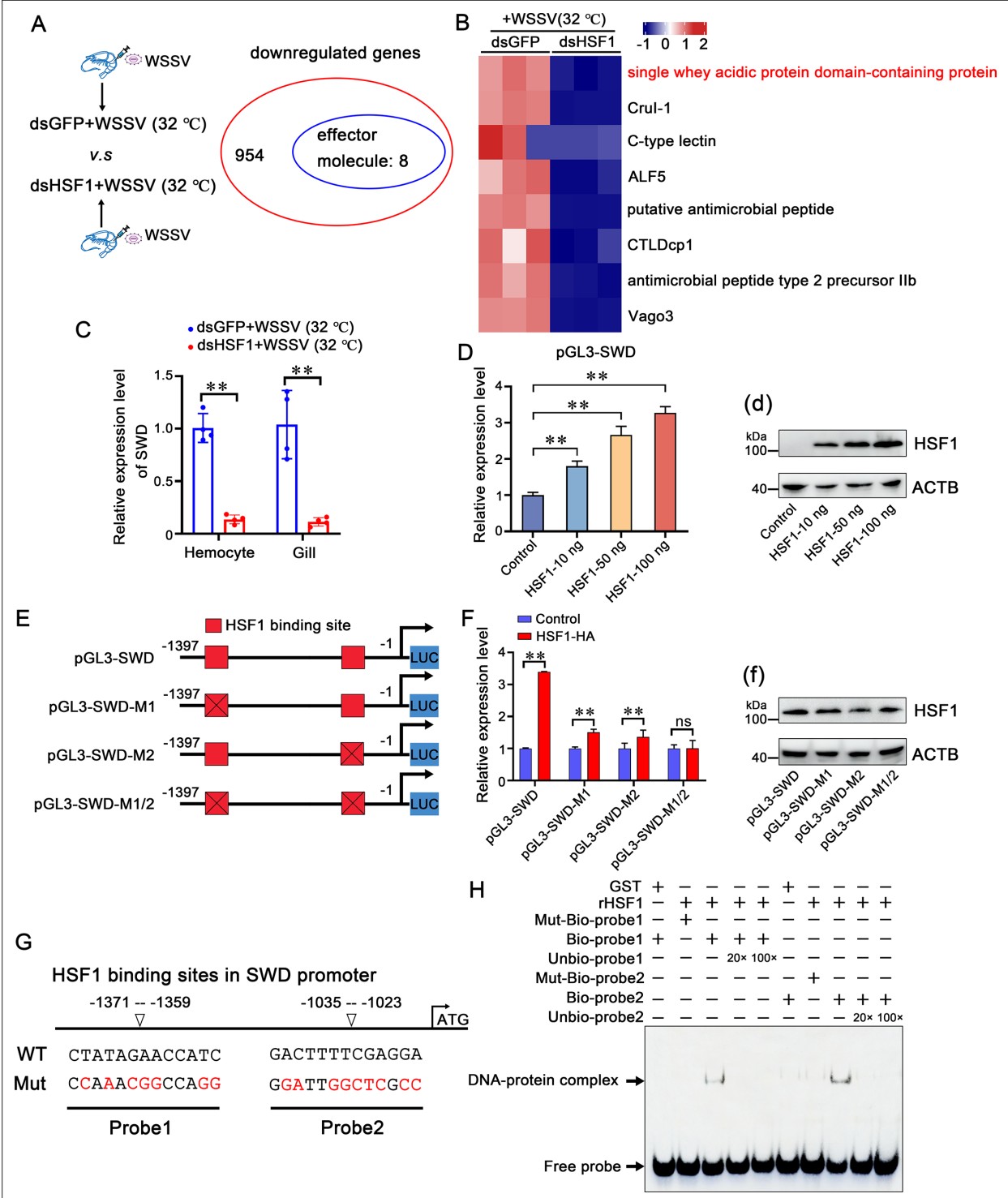

**Figure 5.** SWD is a potential antiviral effector regulated by HSF1. (**A**) Venn diagram showing the downregulated genes by RNA-Seq. RNA-seq was performed 24 hr after WSSV infection (NCBI SRA database accession number PRJNA1110613). (**B**) Heatmap showing the differential expression of downregulated genes encoding potential effector molecules. The color bar indicates the gradient of normalized expression levels. (**C**) mRNA transcription levels of SWD in hemocytes and gills. mRNA transcription levels of SWD in the hemocytes and gills of LvHSF1-silenced shrimp under WSSV challenge. (**D**) Dual-luciferase reporter assays. Dual-luciferase reporter assays were performed to analyze the effects of overexpression of LvHSF1 on the promoter activities of SWD in *Drosophila* S2 cells in a dose-dependent manner. Protein expression of LvHSF1 was detected with anti-HA Ab, with β-actin used as a protein loading control (panel d). (**E**) Schematic diagram of the SWD promoter regions. Schematic diagram of the SWD promoter regions in the luciferase reporter gene constructs. The HSF1 binding motif sites are shown in red rectangles. (**F**) Dual-luciferase reporter assays with mutated HSF1

*Figure 5 continued on next page*

*Figure 5 continued*

binding motifs. Dual-luciferase reporter assays were performed to analyze the effects of overexpression of LvHSF1 on the promoter activities of SWD with mutated HSF1 binding motifs. Protein expression of LvHSF1 was detected with anti-HA Ab, with β-actin used as a protein loading control (panel f). (**G**) Analysis of the SWD promoter. The HSF1 binding site was analyzed using the online JASPAR database. (**H**) EMSA assay. LvHSF1 protein interaction with HSF1 binding sites from the SWD promoter was analyzed in vitro by EMSA assay. Competition assays were performed in the presence of excess unlabeled probes. Statistical significance was calculated using the Student's *t*-test (**p<0.01, *p<0.05). All experiments were conducted with three biological replicates, consistently yielding similar results.

The online version of this article includes the following source data and figure supplement(s) for figure 5:

**Source data 1.** Numerical source data for graphs shown in *Figure 5B, C, D and F*.

**Source data 2.** TIF file with original western blots and boxes indicating the relevant bands shown in *Figure 5d and f*.

**Source data 3.** Original files for western blot analysis displayed in *Figure 5d and f*.

**Figure supplement 1.** SWD conserved sequence alignment in different shrimp species.

**Figure supplement 2.** The expression of effector molecules as determined by qPCR.

**Figure supplement 2—source data 1.** Numerical source data for graphs shown in *Figure 5—figure supplement 2*.

**s**ingle **w**hey acidic protein **d**omain-containing protein, designated SWD. The SWD belongs to type III crustins, which are crustacean cationic cysteine-rich AMPs containing one or two whey acidic protein (WAP) domains at the carboxyl terminus and mainly exert antimicrobial activities (*Figure 5—figure supplement 1*). The detailed regulatory mechanism of SWD against WSSV was unclear. To validate the high-throughput sequencing results, the expression level of SWD and effector molecules was confirmed by qPCR, showing a significant reduction in hemocytes and gills after LvHSF1-knockdown (*Figure 5C*; *Figure 5—figure supplement 2*). These results suggest that SWD may play a pivotal role against WSSV infection regulated by LvHSF1.

To determine whether LvHSF1 regulates the expression of SWD in vitro, a dual-luciferase reporter assay was performed in *Drosophila* S2 cells. The promoter region of SWD was cloned by genome walking, containing two putative conserved HSE motifs located at approximately −1371 to −1359 (HSE-1) and −1035 to −1023 (HSE-2; *Supplementary file 3*). The dual-luciferase reporter assay showed that the promoter of SWD could be regulated by ectopic expression of LvHSF1 in a dose-dependent manner (*Figure 5D and d*). Three reporter plasmids containing mutant promoter regions of SWD with one or no HSE motifs were constructed, named pGL3-SWD-M1, pGL3-SWD-M2, and pGL3-SWD-M1/2, respectively (*Figure 5E*). The reporter assay results showed that the ectopic expression of LvHSF1 could also improve the promoter activity of pGL3-SWD-M1 and pGL3-SWD-M2, whereas the activity of pGL3-SWD-M1/2 was not upregulated (*Figure 5F and f*), suggesting that putative HSE binding motifs play significant roles in LvHSF1-mediated SWD induction.

To further confirm this, EMSA experiments were performed to investigate the direct interaction between LvHSF1 and HSE motifs of SWD. Biotin-labeled probes or mutant probes of the HSE motifs were synthesized (*Figure 5G*). LvHSF1 protein was purified to verify its ability to bind to the HSE motifs (*Figure 6—figure supplement 1A and B*). The results showed that band shifts of protein-DNA complexes were detected when LvHSF1 protein was incubated with biotin-labeled probe1 but not mutant biotin-labeled probe1 (*Figure 5H*). Additionally, the band shifts could be competitively reduced when LvHSF1 protein was incubated with wild-type unbiotinylated probe1 at 20- and 100-fold molar excess (*Figure 5H*). A similar result was observed in the interaction between LvHSF1 protein and the biotin-labeled probe2 (*Figure 5H*). No band shift was observed in the control set (GST), suggesting that the interaction between LvHSF1 protein and each of the HSE binding motifs is specific. More-over, the recombinant protein HSF1 could significantly induce the expression level of SWD in shrimp (*Figure 6—figure supplement 1C*). These results strongly demonstrate that LvHSF1 could regulate the expression of SWD in vitro and in vivo.

## SWD inhibits WSSV replication and interacts with WSSV envelope proteins

To confirm whether SWD is involved in WSSV infection, qPCR was performed to determine the expression profile of SWD. When shrimp were infected with WSSV, the expression of SWD was upregulated at 4 hr (5.66-fold) and 12 hr (5.60-fold), remained high at 36 hr (5.18-fold) and 48 hr (7.90-fold), and slightly downregulated at 72 hr (4.74-fold; *Figure 6A*). To reveal the role of SWD in WSSV infection at

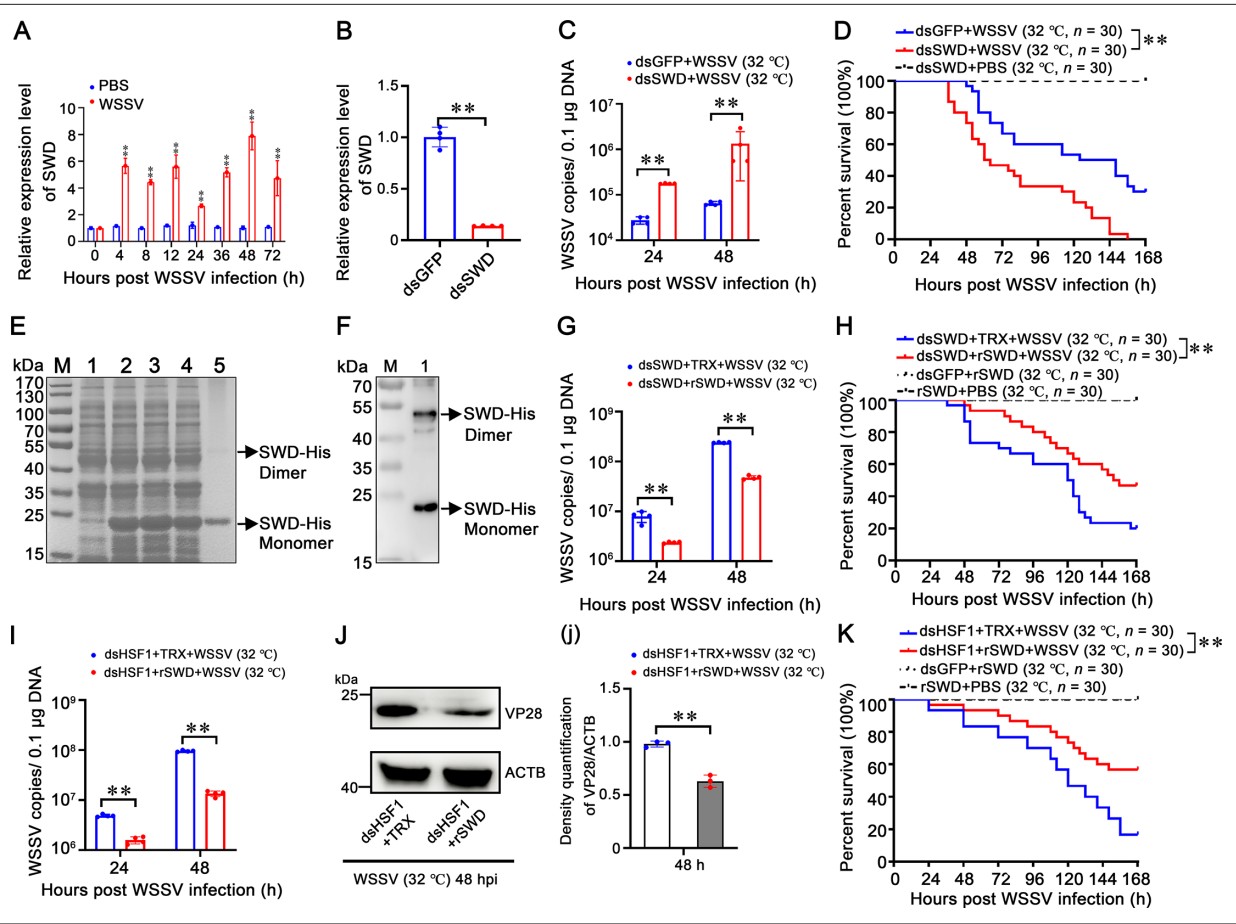

**Figure 6.** *SWD* possessed potent antiviral activities against WSSV. (**A**) Expression of SWD in gills from WSSV-challenged shrimp. (**B**) Silencing efficiency of SWD in gills. Silencing efficiency of SWD in gills 48 hr after WSSV infection. (**C**) Quantity of WSSV copies in gills at high temperature (32 °C). The quantity of WSSV copies in gills at high temperature (32 °C) was detected by absolute quantitative PCR at 24 and 48 hr after WSSV infection. DsGFP was used as a control. (**D**) Survival rates of WSSV-infected shrimp post-SWD knockdown. Survival rates of WSSV-infected shrimp post-SWD knockdown at high temperature (32 °C) were monitored, with recordings made every 4 hr. Statistical analysis was performed using the Kaplan-Meier plot (log-rank $\chi^2$ test). (**E**) SDS-PAGE analysis of recombinant SWD protein expressed in *E. coli*. Line 1: Uninduced *E. coli* transformed with SWD; Line 2: Induced *E. coli* transformed with SWD; Line 3: Supernatant of ultrasonic lysed *E. coli* expressing SWD; Line 4: Precipitate of lysed *E. coli* expressing SWD. Line 5: Purified recombinant SWD protein (black arrow). (**F**) Western blotting of purified rSWD protein. Purified rSWD protein was checked by Western blotting with anti-6×-His Ab. (**G**) Quantity of WSSV copies in shrimp gills following SWD application post-SWD knockdown. The quantity of WSSV copies in shrimp gills following 10 μg SWD application post-SWD knockdown at high temperature (32 °C) was measured in vivo at 24 and 48 hr using absolute quantitative PCR. (**H**) Survival rates post-rSWD application. Shrimp were injected with 10 μg rSWD or control protein, mixed with WSSV inoculum post-SWD knockdown at high temperature (32 °C). Survival rates were recorded every 4 hr. (**I**) Quantity of WSSV copies in shrimp gills following SWD application post-LvHSF1 knockdown. The quantity of WSSV copies in shrimp gills following 10 μg SWD application post-LvHSF1 knockdown at high temperature (32 °C) was measured in vivo at 24 and 48 hr using absolute quantitative PCR. (**J**) Expression of WSSV VP28 protein. Expression of WSSV VP28 protein was detected by Western blotting in gills at high temperature (32 °C) after WSSV infection. Quantification and statistical analysis of three independent repeats were performed using ImageJ (panel j). (**K**) Survival rates post-rSWD application. Shrimp were injected with 10 μg rSWD or control protein, mixed with WSSV inoculum post-LvHSF1 knockdown at high temperature (32 °C). Survival rates were recorded every 4 hr. Statistical significance was calculated using the Student's *t*-test (**p<0.01, *p<0.05). All experiments were conducted with three biological replicates, consistently yielding similar results.

The online version of this article includes the following source data and figure supplement(s) for figure 6:

**Source data 1.** Numerical source data for graphs shown in ***Figure 6A, B, C, D, G, H, I, j and K***.

**Source data 2.** TIF file with original western blots and boxes indicating the relevant bands shown in ***Figure 6F and J***.

**Source data 3.** Original files for western blot analysis displayed in ***Figure 6F and J***.

**Figure supplement 1.** SDS-PAGE and western blotting analyses of the recombinant LvHSF1 protein expressed in *E. coli*.

**Figure supplement 1—source data 1.** Numerical source data for graphs shown in ***Figure 6—figure supplement 1C and D***.

high temperature (32 °C), RNA interference (RNAi) was performed to knock down SWD expression. The mRNA level of SWD was effectively suppressed by specific dsSWD in gills, and the value was downregulated to 0.30-fold of the dsGFP (green fluorescent protein) injection group (as a control; *Figure 6B*). The viral load of the dsSWD group was considerably higher than the control group (*Figure 6C*). Additionally, the survival rate of shrimp in the SWD-silenced group was significantly lower than in the GFP-knockdown group (p<0.01; *Figure 6D*).

To further demonstrate the actual role of SWD in vivo, RNAi experiments coupled with rSWD proteins injection were performed. Recombinant proteins of rSWD and rTrx-His-tag were expressed, purified by Ni-NTA resin, and confirmed by Coomassie staining (*Figure 6E*), and the results were further confirmed by western blot analysis (*Figure 6F*). After the knockdown of endogenous SWD, shrimp samples were treated with WSSV mixed with rSWD or rTrx-His-tag by intramuscular injection at high temperature (32 °C). Consequently, the viral load of those co-injected with rSWD was significantly reduced compared to the control group (p<0.01; *Figure 6G*). Additionally, the survival rates of shrimp co-injected with rSWD were significantly higher than the control group (p<0.01; *Figure 6H*). Therefore, SWD plays a crucial role in the immune defense against WSSV infection at high temperature (32 °C) in vivo.

Next, to verify whether the anti-WSSV function of SWD was regulated by LvHSF1 at high temperature (32 °C), in vivo RNAi-LvHSF1 experiments coupled with rSWD injection were performed. After the knockdown of LvHSF1, shrimp samples were treated with WSSV mixed with rSWD or rTrx-His-tag by intramuscular injection at high temperature (32 °C). The results showed that the viral load of those co-injected with rSWD was significantly reduced compared to the control group (p<0.01; *Figure 6I*). Additionally, VP28 protein levels of those co-injected with rSWD were significantly decreased (*Figure 6J and j*). The survival rates of shrimp co-injected with rSWD were significantly higher than the control group (p<0.01; *Figure 6K*). Moreover, after the knockdown of SWD, shrimp were injected with rLvHSF1 mixed with WSSV. The results showed that the viral load was significantly lower than the control group 48 hr post WSSV infection at high temperature (32 °C) (*Figure 6—figure supplement 1D*). Taken together, these results suggest that the anti-WSSV function of SWD at high temperature (32 °C) is mediated by LvHSF1.

To reveal the possible anti-WSSV mechanism of SWD, pull-down assays were performed to detect whether rSWD protein could interact with WSSV envelope proteins. The main envelope proteins of WSSV, including VP19, VP24, VP26, and VP28 with GST-tags, were from a previous study. In the GST-tagged pull-down assays, rSWD could interact with VP24 and VP26 based on SDS-PAGE gels with Coomassie blue staining (*Figure 7A*), and the result was further confirmed by Western blot analysis with the 6×His antibody (*Figure 7B*). Based on the His-tagged pull-down assays, VP24 and VP26 could precipitate rSWD (*Figure 7C*), and this result was further confirmed by western blot analysis with GST antibody (*Figure 7D*). Additionally, confocal analysis confirmed that SWD colocalized with VP24 (*Figure 7E and e*) and VP26 (*Figure 7F and f*) in *Drosophila* S2 cells. Collectively, these results suggest that SWD could inhibit WSSV replication and bind to VP24 and VP26 of WSSV.

## Conservation of the febrile temperature-inducible HSF1-AMPs axis conferring virus resistance in arthropods

To demonstrate whether HSF1 regulation of AMPs is a conserved defense mechanism induced by elevated temperature in arthropods, experiments were performed in an invertebrate model system (*Drosophila* S2 cells). Literature reports that the temperature range for optimal growth of *Drosophila* S2 cells is 25–28 °C, and S2 cells tolerated a normal growth temperature of 30 °C. Therefore, S2 cells were cultured at normal temperature (27 °C) or high temperature (30 °C) with or without DCV infection. The results showed that S2 cells were adherent to the wall and grew well at 27 °C or 30 °C without DCV infection, and the cytopathic signs of S2 cells were milder at 30 °C than the control group at 27 °C when infected with DCV (*Figure 8—figure supplement 1A*). Additionally, the mRNA and protein expression of DCV was significantly lower at 30 °C than the control group at 27 °C (*Figure 8—figure supplement 1B and C*). These results suggest that elevated temperature could restrict the replication of DCV in *Drosophila* S2 cells.

To determine whether *Drosophila melanogaster* HSF1 (DmHSF1) is related to high temperature (30 °C) and DCV infection in S2 cells, its expression level was also determined. The results showed that the mRNA expression of DmHSF1 was significantly upregulated 4.53-fold and 6.48-fold at

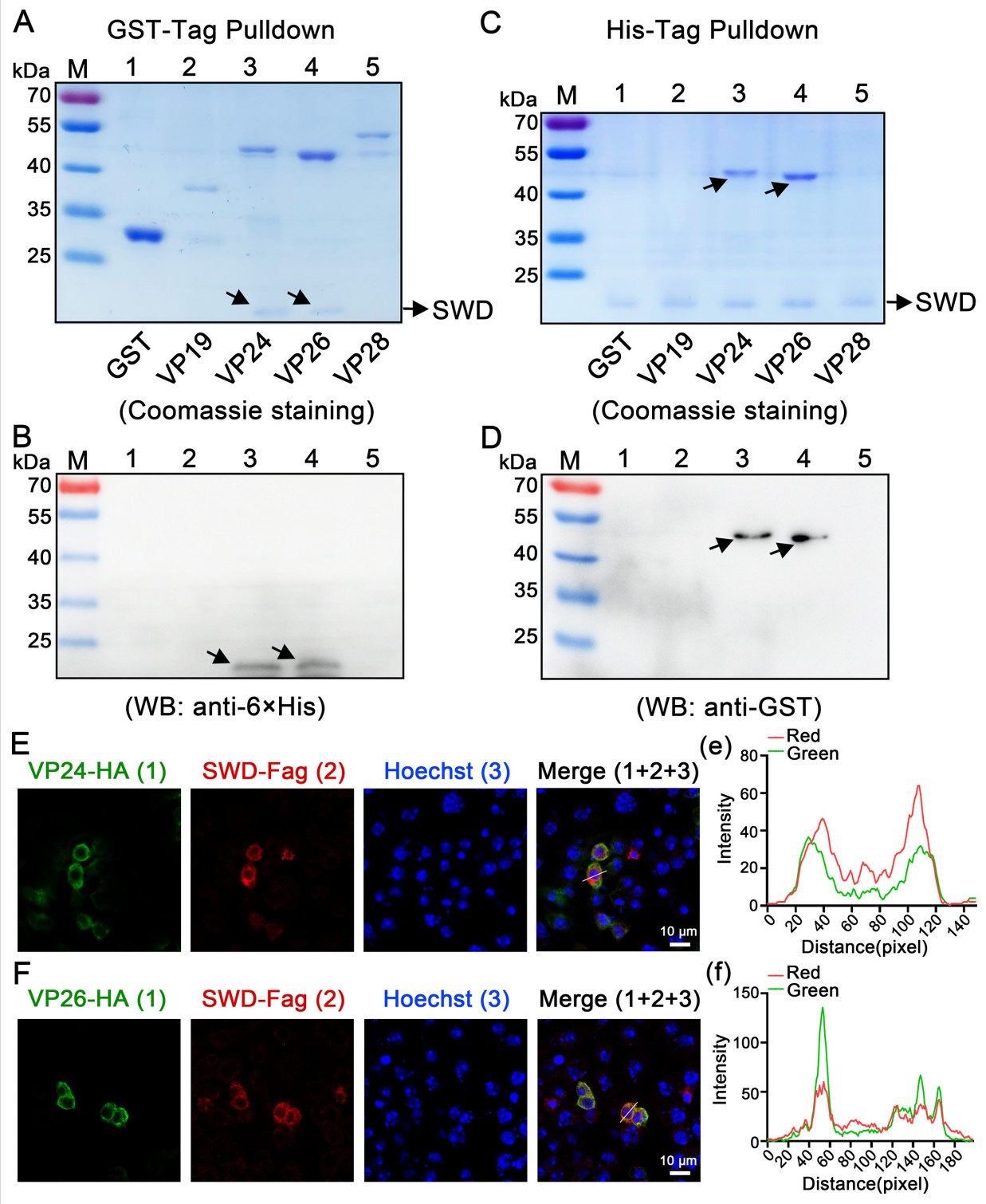

**Figure 7.** SWD interacts with envelope proteins of WSSV. (**A, B**) GST pull-down assay for the detection of the interaction between rSWD with VP19, VP24, VP26, and VP28. The results were shown via staining with Coomassie blue (**A**) or western blotting using 6×-His Ab (**B**). The GST-tag protein was used as a control. (**C, D**) His pull-down assay for the detection of the interaction of rSWD with VP19, VP24, VP26, and VP28 via Coomassie blue staining (**C**) or western blotting using the GST-tag Ab (**D**). The GST tag protein was used as a control. (**E, F**) The colocalization of SWD with VP24 and VP26 in S2 cells, 24 hr post-plasmid transfection. VP24 and VP26 were detected with rabbit anti-HA antibodies and anti-rabbit Alexa Fluor 488, while SWD was identified using anti-Flag antibodies and anti-mouse Alexa Fluor 594. DAPI staining highlighted the nuclei. The scale bar represents 10 μm.

*Figure 7 continued on next page*

Figure 7 continued

(e–f) Quantitative analysis of fluorescence colocalization. Colocalization intensity was quantitatively analyzed, with complete colocalization indicated by overlapping peaks and maxima shifted by less than 20 nm. All experiments were representative of three biological replicates and yielded similar results.

The online version of this article includes the following source data for figure 7:

Source data 1. Numerical source data for graphs shown in *Figure 7e and f*.

Source data 2. TIF file with original western blots and boxes indicating the relevant bands shown in *Figure 7B and D*.

Source data 3. Original files for western blot analysis displayed in *Figure 7B and D*.

high temperature (30 °C) or infected with DCV, respectively (*Figure 8A*). AMPs, especially Attacin A, Cecropins A, Defensin, Metchnikowin, and Drosomycin, have been proven to participate in the innate immunity of *Drosophila* in response to viral infection (*Zhu et al., 2013*). The expression of DmAMPs was explored in response to high temperature (30 °C) and DCV infection. The results showed that Atta, CecA, and Def were significantly upregulated 5.77-fold, 4.55-fold, and 5.40-fold at high temperature (30 °C), respectively (*Figure 8B*). When stimulated by DCV, Atta, CecA, Def, Met, and Drs were remarkably increased 46.84-fold, 123.52-fold, 39.72-fold, 41.14-fold, and 241.40-fold, respectively (*Figure 8C*). These results suggest that DmAMPs play a crucial role in anti-DCV defense at high temperature (30 °C).

Next, DmHSF1's role in *Drosophila* S2 cells during DCV infection at high temperature (30 °C) was investigated. Using the RNAi strategy in S2 cells, it was observed that the transcription levels of DmHSF1 can be downregulated by dsRNA-DmHSF1 with or without DCV infection (*Figure 8D*). After DmHSF1 knockdown, the mRNA and protein expression of DCV was significantly higher at 30 °C than the control group (*Figure 8E and e*). Additionally, Atta, CecA, and Def were significantly downregulated 0.65-fold, 0.43-fold, and 0.56-fold in DmHSF1 silenced, DCV infected S2 cells at high temperature (30 °C), respectively (*Figure 8F*). Furthermore, S2 cells were transfected with pAc5.1a-DmHSF1-HA or pAc5.1a-HA plasmid (as control; *Figure 8—figure supplement 2A*). The results showed that the mRNA and protein expression of DCV was significantly lower at 30 °C than the control group (*Figure 8G and g*). Accordingly, Atta, CecA, and Def were significantly upregulated 4.38-fold, 1.89-fold, and 3.13-fold at high temperature (30 °C), respectively (*Figure 8H*). These results show that high temperature (30 °C) and DCV could induce the expression of DmHSF1 and DmAMPs, with DmHSF1 playing crucial roles in defending against DCV at elevated temperatures. To further verify whether DmHSF1 exerts its anti-DCV function by regulating DmAMPs at high temperature (30 °C), the regulatory roles of DmHSF1 on DmAMPs were investigated through ectopic expression of DmHSF1 and knockdown of DmAMPs during infection. Firstly, the promoter region of DmAtta, DmCecA, and DmDef was cloned, each containing three conserved HSE motifs (*Figure 8I*). Dual-luciferase reporter assays showed that the promoter of DmAtta, DmCecA, and DmDef could be upregulated 12.36-fold, 7.76-fold, and 2.71-fold by DmHSF1, respectively. The mutant HSE motifs promoter of DmAtta, DmCecA, and DmDef could not be regulated by DmHSF1 (*Figure 8J*). Further, due to the expression of Attacin A being the highest DmAMPs induced by DmHSF1, to demonstrate that DmHSF1 exerts its anti-DCV function by regulating DmAMPs, DmAtta was chosen as a representative. After the knockdown of DmAtta (*Figure 8—figure supplement 1B*), S2 cells were transfected with DmHSF1 and then infected with DCV. The results showed that the mRNA and protein expression of DCV was significantly lower at 30 °C than the control group (*Figure 8K and k*). These results suggest that DmHSF1 exerts its anti-DCV function by regulating DmAMPs at high temperature (30 °C).

## Discussion

Temperature is a key factor regulating various biological activities, and for aquatic ectotherms, it significantly affects growth rates and is also a crucial determinant of viral disease outbreaks (*Marcogliese, 2008*). Accumulating evidence has indicated that increased temperature enhances immune competence, resistance to infection, and levels of immune mediators under febrile conditions (*Evans et al., 2015*; *Hasday and Singh, 2000*). High temperatures (32–33°C) have been shown to inhibit the replication of WSSV and reduce mortality in WSSV-infected shrimp (*Sun et al., 2014*). Studies have reported that WSSV-infected shrimp reared at high temperatures (above 31 °C) exhibit increased total hemocyte count (THC), phenoloxidase (PO) activity, and apoptotic index (*Granja et al., 2003*;

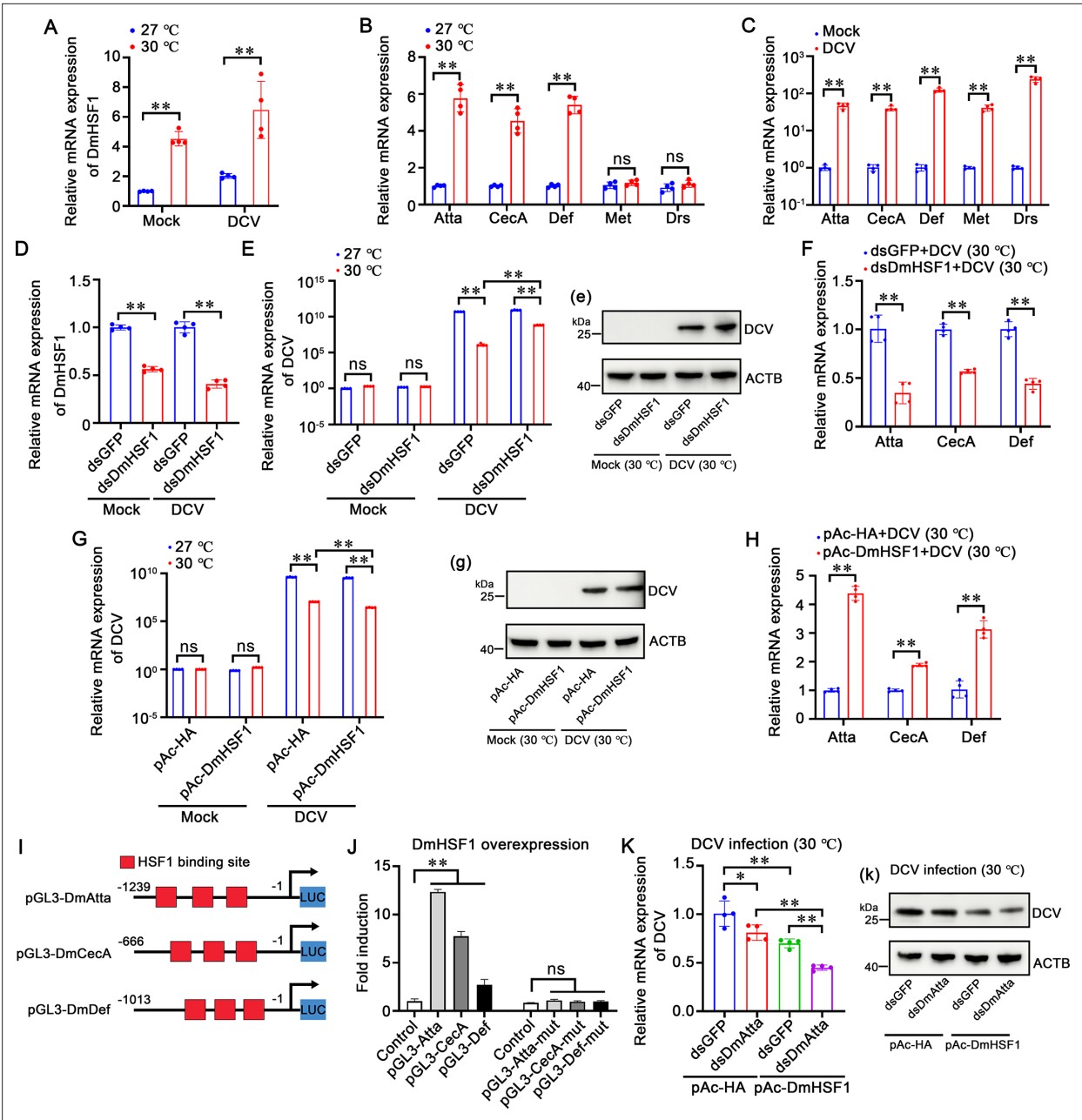

**Figure 8.** High temperature-induced DmHSF1-AMPs axis restricts DCV replication in *Drosophila* S2 cells. (**A**) Transcriptional expression of DmHSF1. Transcriptional expression of DmHSF1 in S2 cells with or without DCV infection at low (27 °C) and high (30 °C) temperatures. (**B**) Transcriptional expression of DmAMPs. Transcriptional expression of DmAMPs in S2 cells at low (27 °C) and high (30 °C) temperatures. (**C**) Transcriptional expression of DmAMPs with or without DCV infection. Transcriptional expression of DmAMPs in S2 cells with or without DCV infection. (**D**) Silencing efficiency of DmHSF1. Silencing efficiency of DmHSF1 in S2 cells 24 hr after dsRNA application with or without DCV infection. (**E**) mRNA expression of DCV. mRNA expression of DCV at low (27 °C) and high (30 °C) temperatures after DmHSF1 knockdown, with dsGFP used as a control. (**e**) Protein expression of DCV at low (27 °C) and high (30 °C) temperatures after DmHSF1 knockdown. (**F**) mRNA expression of DmAMPs. mRNA expression of DmAMPs at low (27 °C) and high (30 °C) temperatures after DmHSF1 knockdown and DCV infection. (**G**) mRNA expression of DCV after DmHSF1 overexpression. mRNA expression of DCV at low (27 °C) and high (30 °C) temperatures after DmHSF1 overexpression. (**g**) Protein expression of DCV at low (27 °C) and high (30 °C) temperatures after DmHSF1 overexpression. (**H**) mRNA expression of DmAMPs after DmHSF1 overexpression. mRNA expression of DmAMPs at low (27 °C) and high (30 °C) temperatures after DmHSF1 overexpression and DCV infection. (**I**) Schematic Diagram of DmAMPs Promoter Regions. Schematic diagram of the DmAMPs promoter regions in the luciferase reporter gene constructs. The HSF1 binding motif sites are shown in red rectangles. (**J**) Dual-luciferase reporter assays. Dual-luciferase reporter assays were performed to analyze the effects of DmHSF1 overexpression on the promoter activities of DmAMPs with or without mutated HSF1 binding motifs. (**K**) mRNA expression of DCV following DmHSF1 overexpression post-DmAtta knockdown. mRNA expression of DCV at high (30 °C) temperatures following DmHSF1 overexpression post-DmAtta knockdown. (**k**) Protein

*Figure 8 continued on next page*

*Figure 8 continued*

expression of DCV at high (30 °C) temperatures following DmHSF1 overexpression post-DmAtta knockdown. Statistical significance was calculated using the Student's *t*-test (**p<0.01, *p<0.05). All experiments were conducted with three biological replicates, consistently yielding similar results.

The online version of this article includes the following source data and figure supplement(s) for figure 8:

**Source data 1.** Numerical source data for graphs shown in ***Figure 8A, B, C, D, E, F, G, H, J and K***.

**Source data 2.** TIF file with original western blots and boxes indicating the relevant bands shown in ***Figure 8e, g and k***.

**Source data 3.** Original files for western blot analysis displayed in ***Figure 8e, g and k***.

**Figure supplement 1.** High temperature restricts DCV replication in *Drosophila* S2 cells.

**Figure supplement 1—source data 1.** Numerical source data for graphs shown in ***Figure 8—figure supplement 1B***.

**Figure supplement 1—source data 2.** TIF file with original western blots and boxes indicating the relevant bands shown in ***Figure 1C***.

**Figure supplement 1—source data 3.** Original files for western blot analysis displayed in ***Figure 1C***.

**Figure supplement 2.** Transfection in S2 cells.

**Figure supplement 2—source data 1.** Numerical source data for graphs shown in ***Figure 8—figure supplement 2B***.

*Wu et al., 2015*). However, the mechanism contributing to enhanced pathogen resistance in shrimp at high temperatures remains unclear. In this study, our findings revealed the molecular mechanism through which the HSF-AMPs axis mediates host resistance to viruses induced by febrile temperature. Specifically, AMPs are a newly documented type of effector protein targeted by HSF that mediate the host antiviral response beyond the classical HSR.

Regulating body temperature, particularly increasing it (fever), is one of the primary defense mechanisms in homeotherms against infections (*Hasday et al., 2014*). Ectotherms as diverse as reptiles, fish, insects, and crustaceans also raise their core temperature during infection through behavioral regulation, seeking warmer environments—a phenomenon known as behavioral fever (*Hasday et al., 2014*). This feature, preserved in endothermic mammals and birds, strongly suggests that febrile temperatures confer a survival advantage (*Bicego et al., 2007*). In 1975, Kluger and colleagues published the first direct experimental study demonstrating the benefits of elevated temperature for infected animals (*Kluger et al., 1975*). In this study, *dipsosaurus dorsalis*, being poikilothermic animals, were moved from an environment of 36–40°C after bacterial infection, resulting in an increase in survival rates from 25% to 65% (*Kluger et al., 1975*). Similar results have since been observed in other poikilotherms and some homeotherms. For instance, carp infected with Cyprinid herpesvirus 3 (CyHV-3) move from lower temperatures of 24–28°C to 32 °C, where their survival rate reaches 100%, and viral load is significantly reduced (*Rakus et al., 2017*). Behavioral fever responses to microbial infections have also been reported in aquatic invertebrates such as freshwater crayfish (*Cambarus bartoni*), American lobster (*Homarus americanus*), pink shrimp (*Penaeus duorarum*), horseshoe crab (*Limulus polyphemus*), and *Penaeus vannamei* (*Casterlin and Reynolds, 1978*; *Casterlin and Reynolds, 1979*; *Casterlin and Reynolds, 1977*; *Rakhshaninejad et al., 2023*). Despite these observations, the antimicrobial immune mechanisms induced by febrile temperatures remain unclear. The prevailing view that febrile temperatures benefit the host primarily suggests that febrile response enhances innate and adaptive immune mechanisms in homeotherms (*Hasday et al., 2014*). Our findings in the present study further support the enhancement of innate immune mechanisms by febrile temperature in ectotherms such as shrimp and *Drosophila* S2 cells. Additionally, elevated temperatures can cause some pathogenic microorganisms to lose their optimal growth conditions. However, some microorganisms have evolved strategies to exploit fever to enhance their invasion and proliferation (*Freitas Lione et al., 2010*; *Loh et al., 2013*). Paradoxically, this further underscores the benefits of fever for the host, as effective immune strategies exert evolutionary pressure on pathogens, driving them to evolve counterstrategies to hijack or block the febrile response.

HSFs are a type of evolutionarily conserved transcription factors initially known for regulating the HSR (*Anckar and Sistonen, 2011*). However, accumulating evidence indicates multiple additional functions beyond the activation of HSPs, including roles in metabolism, immunity, disease, organismal aging, and longevity (*Barna et al., 2018*; *Lee and Lee, 2022*). Additionally, studies also reveal the role of HSF1 in hormonal responses, which are evolved to maintain cellular homeostasis and ensure the survival of organisms under changing environmental conditions (*Kumsta et al., 2017*; *Lee and Lee, 2022*). HSF1 acts through a regulated upstream promoter element called the HSE. HSEs are highly

conserved, including the inverted repeats of the pentameric sequence nGAAn (*Amin et al., 1988*). The type of HSEs found in the proximal promoter regions of HSP genes comprises at least three contiguous inverted repeats consisting of nTTCnnGAAnnTTCn (*Xiao and Lis, 1988*). The discovery of novel HSF1 target genes not involved in the HSR suggests that HSEs may be present in multiple genes beyond HSPs (*Barna et al., 2018*). For instance, HSF1 can bind to the promoters of CUP1, BTN2, SIS1, SGT2, and SSA3 genes, upregulating their transcriptional activity to regulate oxidative stress (*Morano et al., 2012*). HSF1 also regulates the hypothalamic-specific expression of the heat-inducible ion channel gene TRPV1, suggesting a protective effect during fever (*Fan-xin et al., 2012*). In the present study, we showed that a newly identified AMP, SWD, is specifically regulated by LvHSF1 via two HSE motifs with the typical nGAAn sequence. Additionally, the promoters of DmAMPs, such as Atta, CecA, and Def, also contain several HSE motifs and are regulated by DmHSF1. Previous studies showed that the upstream sequence of CrustinPm5 contains a putative promoter with an HSE motif, and the expression of CrustinPm5 is upregulated upon heat treatment (*Vatanavicharn et al., 2009*). However, there is currently no evidence that HSF1 directly binds to the HSE motif of AMPs in invertebrates. Notably, our work is the first to show that HSF1 directly binds to the HSE motifs of AMPs in shrimp and *Drosophila* S2 cells, suggesting this may be a conserved regulatory mechanism in arthropods.

Although febrile temperature is generally believed to enhance host immunity, the role of HSFs in viral infections is complex and varies across different species and types of viral infections (*Reyes et al., 2022*). In vertebrates, HSF1 has been reported to participate in the transcription of HIV genes and their reactivation from latency (*Pan et al., 2016*). Additionally, after orthopoxvirus infection, HSF1 is phosphorylated, translocated into the nucleus, and increases the transcription of HSF1 target genes, benefiting the virus by extending the viral genome (*Filone et al., 2014*). Conversely, stress-independent activation of HSF1 can reduce the quantity and infectivity of HIV copies in a lymphoblastic cell line (*Nekongo et al., 2020*). Another study reported that HSF1 could enable apoptosis-stimulating protein to reduce autophagy in hepatocytes and inhibit HBV replication (*Wang et al., 2021*). In invertebrates, DmHSF1-deficient adult flies are hypersensitive to DCV infection, although the underlying mechanism has not yet been uncovered (*Merkling et al., 2015*). HSF1 is also required for tolerance against several pathogenic agents in *Caenorhabditis elegans* (*Singh and Aballay, 2006*). Similarly, the present study showed that HSF1 plays a crucial role in antiviral immunity against WSSV infection, with knockdown of HSF1 in vivo leading to higher virus copies at low temperature (25 °C) and high temperature (32 °C), and increased susceptibility to WSSV infection at high temperature (32 °C). Notably, a recent study showed that ammonia stress-induced MjHSF1 upregulates the expression of negative regulators that inhibit the production and function of interferon analogs, enhancing WSSV infection in shrimp *Marsupenaeus japonicus* (*Wang et al., 2024*). This may be because ammonia stress suppresses host immunity and increases sensitivity to pathogenic infection in aquaculture (*Weihrauch et al., 2018*). In fact, previous studies have shown that the optimal water temperature for *M. japonicus* growth ranges from 25°C to 32 °C, and a high temperature of 31 °C can reduce mortality of WSSV-infected *M. japonicus* without increasing host immune response activity (*You et al., 2010*). In contrast, the tolerance temperature for *L. vannamei* growth ranges from 7.5°C to 42 °C, and our findings showed that elevated temperature can effectively enhance the host's response to WSSV infection (*Millard et al., 2021*). Evidence also supports the involvement of HSPs in WSSV tolerance at elevated temperatures, with HSP70 playing a crucial role in inhibiting WSSV replication at high temperatures (32 °C; *Lin et al., 2011*; *Sun et al., 2014*). These findings suggest that the mechanism of elevated temperature protection against WSSV infection differs between *M. japonicus* and *L. vannamei*. Therefore, our findings, together with those of other studies, suggest that the benefit of HSF1 can be attributed to either the host or the pathogen, depending on the nature and context of the host-virus-environment interaction.

AMPs are pivotal components of innate immunity in invertebrates, serving as the first line of defense against invading pathogens, including bacteria, fungi, and viruses (*Brogden, 2005*). Typically, the transcription of AMPs is regulated by the classical NF-$\kappa$B and JAK/STAT pathways in invertebrates (*Carpenter and O'Neill, 2024*). However, our results revealed that AMPs can also be regulated by an alternative pathway mediated by HSF1 in shrimp and *Drosophila* S2 cells. In crustaceans, various AMP families, including penaeidins, anti-lipopolysaccharide factors, crustins, and stylicins, have been identified (*Matos and Rosa, 2022*). Among these, the newly identified SWD belongs to type III crustins, characterized by a single WAP domain (SWD) and a short N-terminal region. The most probable

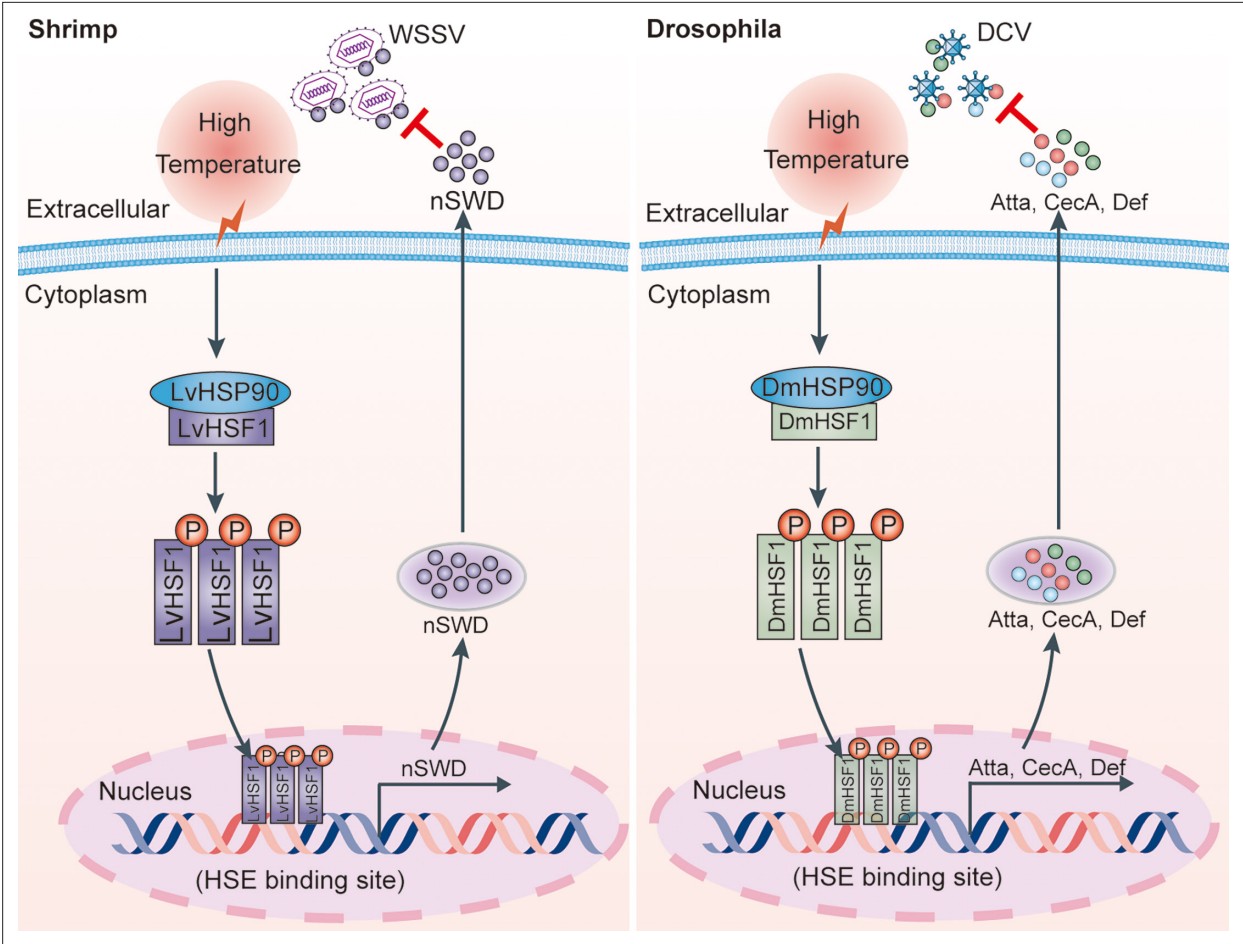

**Figure 9.** Model for the HSF1-AMPs-mediated high temperature-induced resistance to viruses in arthropods. Elevated temperature induces a robust expression of LvHSF1, which in turn specifically induces the expression of the antimicrobial peptide (SWD) in shrimp. The SWD directly binds to WSSV envelope proteins and inhibits WSSV replication (*left panel*). Additionally, elevated temperature induces the expression of DmHSF1, which upregulates the expression of Atta, CecA, and Def in *Drosophila* S2 cells, subsequently restricting the replication of DCV (*right panel*). These findings highlight the roles of HSF1 beyond the classical heat shock response, mediating the thermal regulation of immunity and facilitating the innate immune system's response against viruses.

anti-WSSV mechanism of SWD is its direct interaction with WSSV envelope proteins VP24 and VP26, potentially inhibiting viral entry into target cells. VP24 has been identified as a WSSV receptor-binding protein, interacting with the cellular receptor pIgR to mediate WSSV entry into host cells (**Niu et al., 2019**). VP26 has been reported to bind to SNAP29, inhibiting SNARE complex assembly and autophagic degradation (**Liu et al., 2024**). Additionally, WSSV VP24, VP26, and VP28 form a complex termed an 'infectosome', which is crucial for WSSV infectivity (**Li et al., 2015**). Therefore, the binding of SWD to WSSV envelope proteins VP24 and VP26 could disrupt WSSV integrity and attenuate viral entry into shrimp cells.

In conclusion, our study demonstrates a conserved antiviral febrile response, where the elevated temperature-induced HSF1-AMP axis plays a crucial role in enhancing host resistance to viruses in shrimp and *Drosophila* S2 cells (**Figure 9**). Specifically, febrile temperature-induced HSF1 induces the expression of AMPs, which directly bind to WSSV envelope proteins and block viral entry. Additionally, high temperature restricts DCV replication in the invertebrate model system *Drosophila* S2 cells, also mediated by the HSF1-AMPs cascade. These findings highlight previously unforeseen roles of HSF1 beyond the traditional HSR, revealing its involvement in thermal regulation of immunity and the innate immune system's response to thermal stress. Moreover, this study provides new avenues for managing viral infections in arthropods, such as implementing varied pond depths for temperature stratification as a cost-effective disease prevention method.

## Materials and methods

### Animal and pathogens

Healthy specimens of *L. vannamei*, averaging 5 g each, were utilized for our experiments. These specific pathogen-free (SPF) shrimp were sourced from Hisenor Company's shrimp farm in Maoming, Guangdong Province, China. The shrimp were maintained in a recirculating water tank system filled with air-pumped seawater at a salinity of 25‰ and a constant temperature of 25°C. They were fed thrice daily to satiation with a commercial shrimp diet provided by HAID Group, Guangzhou, China. The inoculum for the WSSV, Chinese strain (AF332093), was prepared following previously described methodologies (*Li et al., 2018*).

WSSV was extracted from the muscle tissue of WSSV-infected shrimp and subsequently stored at –80°C. Prior to injection, the muscle tissue was homogenized to prepare the WSSV inoculum, containing approximately $1 \times 10^5$ copies in 50 µl of Phosphate-Buffered Saline (PBS; 140 mM NaCl, 2.7 mM KCl, 10 mM $Na_2HPO_4$, 1.8 mM $KH_2PO_4$, pH 7.4). For the pathogenic challenge experiments, each shrimp was administered an intraperitoneal injection of 50 µl of the WSSV solution into the second abdominal segment. Injections were performed using a 1 ml syringe (Becton Dickinson, Shanghai, China).

DCV were kindly provided by Professor Jiahuai Han's Lab. The DCV was amplified in the S2 cells. After 3 days of infection, culture medium was collected and centrifuged at 4°C for 20 min ($2000 \times g$), then the supernatant was collected and aliquots were frozen at –80°C.

### Cells and cell assays

*Drosophila* S2 cells (ATCC CRL-1963) were cultured at 28 °C in Schneider's Insect Medium (Sigma-Aldrich, St. Louis, MO, USA) supplemented with 10% fetal bovine serum (Gibco, Grand Island, NY). Cell line identity was authenticated based on supplier information and characteristic morphology. STR profiling is not applicable to insect cell lines. Cells were routinely tested and confirmed to be free of mycoplasma contamination.

To knockdown *DmHSF1* in vitro, $8 \times 10^6$ S2 were transfected with 15 µg dsRNA in a six-well plate by using FuGENE HD Transfection Reagent (Promega Shanghai, China), and transfected cells were incubated at 27°C for 3 days before the transfection and DCV infection assays according to a previously published method (*Yang et al., 2018*).

For temperature assays, S2 cells were seeded into six-well plates and then incubated at 27°C or 30°C for 24 hr. The transcription of DCV and DmHSF1 was detected by qPCR. Besides, the same numbers of dsRNA-treated cells were infected with DCV at a multiplicity of infection (MOI) of 10 in a six-well plate at 27°C or 30°C for 24 hr and replaced with fresh medium, and the total RNA of S2 cells was extracted for qPCR analysis.

### Transcriptomic analysis

Two groups of *L. vannamei* shrimp, Group W and Group TW, each comprising 30 individuals, were initially cultured at 25°C. For Group W, the shrimp were injected with WSSV and continuously maintained at 25°C. In contrast, Group TW shrimp were also injected with WSSV, but the culture temperature was raised to 32°C within 3 hr at 24 hpi. At 12 hptr, gill samples from eight shrimp in both Group W and Group TW were collected. These samples were immediately preserved in liquid nitrogen for subsequent analysis. Transcriptomic sequencing was performed by LianChuan Biotech (Hangzhou, China). The FPKM value was used to represent gene expression levels in each sample. The differential expression between the two groups, each with three biological replicates, was determined using DESeq2 software, accepting criteria of log2 (Fold Change) ≥1 and an adjusted p-value ≤0.05. Gene annotations were provided for the *L. vannamei* shrimp genome (*Zhang et al., 2019*). OmicShare tools (https://www.omicshare.com/tools) were used to predict potential transcription factors and generate a heatmap.

To identify the genes whose expression was regulated by *LvHSF1*, healthy shrimp were administered an intramuscular injection of 10 µg dsRNA targeting the *LvHSF1* gene or *GFP* as a control. After 48 hr, shrimp were injected with $1 \times 10^5$ copies of WSSV. At 24 hr post WSSV challenge, gill samples from eight shrimp in both *dsHSF1* and *dsGFP* were collected. These samples were immediately preserved in liquid nitrogen for subsequent analysis. The sequencing and heatmap were performed similarly to the above procedure.

## RNA extraction, cDNA synthesis, and DNA and protein extraction

Total RNA was extracted from different tissues of shrimp using the Eastep Super Total RNA Extraction Kit (Promega, Shanghai, China). The genomic DNA of shrimp tissues was extracted using a genomic DNA extraction kit (Vazyme, Nanjing, China). First-strand cDNA synthesis was performed using a cDNA synthesis kit (Accurate Biology, Hunan, China), according to the manufacturer's instructions. Protein samples from gills and hemocytes were homogenized separately in IP lysis buffer (25 mM Tris-HCl pH 7.4, 150 mM NaCl, 1 mM EDTA, 1% NP-40, 5% glycerin; Thermo Fisher Scientific, Shanghai, China) with a protease and phosphatase inhibitor cocktail (Merck, Kenilworth, NJ, USA), and then centrifuged at 12,000×$g$ for 10 min at 4°C to collect the supernatant for further analysis.

## Tissue expression and immune challenge analysis by quantitative real-time PCR (qPCR)

qPCR assays were performed to assess the mRNA levels in tissue expression distribution, immune stimulation, or the in vivo RNAi experiments. Expression levels of *HSF1* (GenBank accession No. AHI13794), *SWD* (GenBank accession No. PP786678), and *VP28* (GenBank accession No. NP_477943.1) were detected using LightCycler480 System (Roche, Basel, Germany) in a final reaction volume of 10 μl, which was comprised of 1 μl of 1: 10 cDNA diluted with ddH$_2$O, 5 μl of SYBR Green Pro Taq HS Mix (Accurate Biology, Hunan, China) and 250 nM of specific primers (*Supplementary file 1*).

For tissue distribution, shrimp were cultured at low temperature (25°C) and high temperature (32°C). Shrimp tissues of epithelium, hemocyte, stomach, nerves, eyestalk, gill, antenna, hepatopancreases, muscle, heart, intestine, and pleopod were sampled and pooled from 15 shrimp. For immune stimulation, the treated group was injected with 5 μg polyinosinic-polycytidylic acid (Poly (I: C)) or 50 μl WSSV (~1 × 10$^5$ copies), and the control group was injected with PBS solution. Hemocyte, gill, and intestine of challenged shrimps were collected at 0, 4, 8, 12, 24, 36, 48, and 72 hr after injection. The cycling program was performed as described previously (*Xiao et al., 2021*). The expression of each gene was calculated using the Livak (2$^{-\Delta\Delta CT}$) method after normalization to *EF-1α* (GenBank accession No. GU136229) and *β-Actin* (GenBank accession No. AF186250). Primer sequences are listed in *Supplementary file 1*.

The transcription of *DCV*, *DmHSF1* (GenBank accession No. NM_057227.5), and *DmAMPs* was also detected by quantitative real-time PCR. The expression of each gene was calculated using the Livak (2$^{-\Delta\Delta CT}$) method after normalization to *DmRpL32* (GenBank accession No. NP_733340.1). Primer sequences are listed in *Supplementary file 1*.

## Detection of viral loads by absolute quantitative PCR

To measure viral titers in shrimp, absolute quantitative PCR (ab-qPCR) was employed. This process utilized a set of primers, wsv069 (WSSV32678-F/WSSV32753-R), targeting a single-copy gene of the WSSV, along with a Taq-Man fluorogenic probe (WSSV32706), as detailed in previous studies. Gill tissue samples were collected from WSSV-infected shrimp. DNA extraction from these samples was conducted using previously described methods (*Fu et al., 2022*). The concentration of WSSV genome copies was quantified by ab-qPCR, using the specific primers WSSV32678-F/WSSV32753-R and the TaqMan fluorogenic probe, as listed in *Supplementary file 1*. To ensure accuracy, each shrimp sample underwent three replicates of ab-qPCR. The number of WSSV genome copies was calculated and normalized against 0.1 μg of shrimp tissue DNA.

## Recombinant protein expression and purification

The coding sequences of SWD (without the signal peptide, 1–26 amino acids) were amplified by PCR using corresponding primers (*Supplementary file 1*) and subcloned into pET-32a (+) plasmid (Merck Millipore, Darmstadt, Germany). After being confirmed by sequencing, the recombinant plasmid was transferred into *E. coli* Rosetta (DE3) cells (TransGen Biotech, Beijing, China). Then, positive clones harboring the desired fragment were selected for inducing expression. After 4 hr of induction with 0.1 mM Isopropyl β-D-Thiogalactoside (IPTG) at 30°C, cells were pelleted by centrifugation and sonicated for 30 min on ice water. The supernatant from the sonicated proteins was purified by using Ni-NTA agarose (Qiagen, Düsseldorf, Germany) according to the manufacturer's instructions. The rTrx-His-tag, GST, and HSF1-GST proteins were induced and purified in the same way. The purified proteins were checked by Coomassie staining or western blot analysis. The concentration of the

purified proteins was determined using a BCA protein assay kit (Beyotime Biotechnology, Shanghai, China).

## SDS-PAGE and western blotting

Western blotting was performed to evaluate the protein levels of LvHSF1, VP28, DCV, the purified SWD, rTrx-His-tag, and LvHSF1-GST. And then the proteins were separated on 12.5% SDS-PAGE gels and then transferred to polyvinylidene difluoride (PVDF) membranes (Merck Millipore, Darmstadt, Germany). After blocking with 5% nonfat milk diluted in Tris Buffered Saline with Tween 20 (TBST) buffer (150 mM NaCl, 0.1% Tween-20, 50 mM Tris-HCl, pH 8.0) for 1 hr, the membrane was incubated with 1: 1000 polyclonal rabbit anti-LvHSF1 (produced in Gene Create, Wuhan, China), 1: 1000 polyclonal rabbit anti-VP28 (produced in Abmart, Shanghai, China), 1: 1000 polyclonal rabbit anti-DCV capsid polyprotein (Abcam, Cambridge, UK), 1:1000 mouse anti-6×His (Sigma, St Louis, MO, USA) or 1:1000 mouse anti-GST (TransGen Biotech, Beijing, China) for 2 hr at 25 °C. The PVDF membranes were washed three times with TBST and then incubated with 1:5000 goat anti-rabbit IgG (H+L) HRP or 1:5000 goat anti-mouse IgG (H+L) HRP secondary antibody (Promega, Shanghai, China) for 1 hr. Membranes were developed using an enhanced chemiluminescent blotting substrate (Thermo Fisher Scientific, Waltham, MA, USA), and the chemiluminescent signal was detected using the 5200 Chemi-luminescence Imaging System (Thermo Fisher Scientific, Waltham, MA, USA).

## Knockdown of specific genes by RNA interference (RNAi)

The T7 RiboMAX Express RNAi System kit (Promega, Shanghai, China) was employed for the synthesis of double-stranded RNA (dsRNA) targeting *HSF1*, *SWD,* and *GFP*. The specific primers used for dsRNA synthesis are listed in *Supplementary file 1*. The integrity and quality of the synthesized dsRNA were verified using 1.5% agarose gel electrophoresis and quantified with a NanoDrop 2000 spectropho-tometer (Thermo Scientific, Shanghai, China). For the experimental treatments, shrimp were injected with 2 µg/g of body weight of the respective dsRNAs, dissolved in 50 µl of PBS. Control groups received injections of GFP dsRNA and PBS. Hemocytes and gills were collected from the shrimp 48 hr post-dsRNA injection. Total RNA was extracted from the collected tissue samples. The efficiency of RNAi was evaluated by qPCR using the corresponding primers for each gene. This assessment aimed to confirm the knockdown of the targeted genes in the experimental groups compared to the controls as described previously (*Li et al., 2024*).

## Survival assays

Healthy shrimp were divided into two groups (n=30 each) and received an intramuscular injection of 10 µg dsRNA solution (HSF1 dsRNA or GFP dsRNA diluted in PBS). After 48 hr post-dsRNA injection, the shrimp in each group were injected again with $1 \times 10^5$ copies of WSSV particles diluted in 50 µl PBS. The first group was maintained at low temperature (25°C), the second at a high temperature (32°C). The survival rates of each group were recorded at 4 hr intervals.

To knock down the expression of SWD, healthy shrimp were administered an intramuscular injection of 10 µg dsRNA targeting the *SWD* gene or PBS as a control. After 48 hr, shrimp were injected with $1 \times 10^5$ copies of WSSV, with a parallel group receiving a mock challenge with PBS (n=30). To evaluate the effect of recombinant rSWD on WSSV replication in vivo, rescue experiments were conducted. Shrimp (n=30) were co-injected with 10 µg rSWD or rTRX along with the WSSV inoculum. The survival rates of each group were recorded at 4 hr intervals. Differences in survival between groups were analyzed using the Mantel-Cox (log-rank $\chi^2$ test) method, employing GraphPad Prism software (GraphPad Software, La Jolla, CA, USA).

## Dual-luciferase reporter assays

The partial promoter sequence with 2000 bp upstream of SWD was cloned using the specific primers (*Supplementary file 1*) and then linked into pGL3-Basic (Promega, Shanghai, China) to generate pGL3-SWD (reporter plasmid). The vector pGL3-SWD-M1, pGL3-SWD-M2, and pGL3-SWD-M1/2 with mutant of the HSF1 binding motif was also generated, respectively. *Drosophila* S2 cell was used instead in order to detect the effects of HSF1 on the promoters of SWD. For dual-luciferase reporter assays, S2 cells were plated into a 96-well plate, and 12 hr later, the cells of each well were transfected with 0.05 µg of firefly luciferase reporter gene plasmids, 0.001 µg pRL-TK renilla luciferase plasmid

(Promega, Shanghai, China), or 0.05 µg protein expression plasmids or empty pAc5.1A plasmids (as controls) using the Fugene HD Transfection Reagent (Promega, Shanghai, China) according to the user manual. Forty-eight hours post-transfection, the dual-luciferase reporter assays were performed in order to calculate the relative ratios of firefly and renilla luciferase activities using the Dual-Glo Luciferase Assay System kit (Promega, Shanghai, China), according to the manufacturer's instructions. All experiments were repeated six times.

To explore the regulatory activity of DmHSF1 on the DmAMPs (Atta, CecA, and Def), we constructed the vector pGL3-Atta, pGL3-CecA, and pGL3-Def. And the vector pGL3-Atta, pGL3-CecA, and pGL3-Def with mutant of the HSF1 binding motif were also generated, respectively. The dual-luciferase reporter assays were conducted in the same way.

## EMSA assay

EMSA was performed using a Light Shift Chemiluminescent EMSA kit (Thermo Fisher Scientific, Waltham, MA, USA) according to a previously published method (*Wang et al., 2020*). Briefly, the biotin-labeled or unbiotin-labeled probes were designed using the *HSF1* binding motif sequence (5'-CTATAGAACCATC-3' and 5'- GACTTTTCGAGGA –3'). The mutant probe was designed via mutating the HSF1 binding motif sequence. All of the probes were synthesized by Ruibo Technologies (Guangzhou, China), and sequences are listed in *Supplementary file 1*. In brief, the recombinant proteins (10 µg) of HSF1 were incubated with 20 fmol of the probes for the binding reactions. The reactions were separated on a 5% native PAGE gel, transferred to positively charged nylon membranes (Roche, Germany), and cross-linked by UV light. Then the biotin-labeled DNA on the membrane was detected by chemiluminescence and developed on X-ray film, followed by enhanced chemiluminescence (ECL) visualization (Tanon, Shanghai, China).

## Pull-down assays

Pull-down assays were performed to explore whether the recombinant rSWD could interact with the main envelope proteins of WSSV (VP19, VP24, VP26, and VP28). The recombinant GST-tagged VP19, VP24, VP26, and VP28 were obtained from our previous studies (*Xiao et al., 2020*). For GST pull-down assays, 5 µg rSWD was incubated with 5 µg of GST-tagged WSSV protein solution at 4°C for 3 hr by agitation. Subsequently, 20 µl of the GST resin was added and the agitation continued for another 2 hr. The resin was washed four times with PBS. Finally, the resins were re-suspended in 50 µl of the SDS-PAGE sample buffer, boiled, and analyzed by western blot analysis using 6×His antibody. For His pull-down assays, 5 µg rSWD was incubated with 5 µg of GST-tagged WSSV protein solution at 4°C for 3 hr by agitation. Subsequently, 20 µl of the Ni-NTA binding resin was added and the agitation continued for another 2 hr. The resin was washed four times with PBS. Finally, the resins were re-suspended in 50 µl of the SDS-PAGE sample buffer, boiled, and analyzed by western blot analysis using GST-tag antibody.

## Immunocytochemical staining

Immunocytochemical staining was used to observe the co-localization of SWD and WSSV VP proteins. *Drosophila* S2 cells were seeded onto poly-L-lysine-treated glass cover slips in a 24-well plate with approximately 40% confluent. The S2 cells of each well were transfected with 0.5 µg pAc5.1-SWD-Flag, pAc5.1-VP24-HA, or pAc5.1-VP26-HA using the MIK-3000 Transfection Reagent (MIKX, Shenzhen, China). Thirty-six hours post-plasmid transfection, cell culture medium was removed, and cells were washed with PBS three times. Cells were then fixed with 4% paraformaldehyde at 25 °C for 15 min following with methanol permeabilization at –20 °C for 10 min. After washing sides for three times, cells were blocked with 3% bovine serum albumin (dissolved in PBS) for 60 min at 25 °C and then incubated with rabbit anti-HA antibody (CST, 1:300 diluted in 2% BSA) and mouse anti-Flag antibody (CST, 1:300 diluted in 2% BSA). The slides were then washed with PBS three times and then incubated with anti-rabbit IgG (H+L) Alexa Fluor 488 (CST, 1:1000 diluted in 2% BSA) and anti-mouse IgG (H+L) Alexa Fluor 594 (CST, 1:1000 diluted in 2% BSA) for 60 min at 25 °C in the dark. The cell nuclei were stained with Hoechst (Beyotime, Shanghai, China; cat. no. C1002) for 5 min. Finally, fluorescence was visualized on a confocal laser scanning microscope (Leica TCS-SP8, Wetzlar, Germany).

## Statistical analysis

All data were presented as means ± SD. Student's $t$ test was used to calculate the comparisons between groups of numerical data. For survival rates, data were subjected to statistical analysis using GraphPad Prism software (GraphPad, San Diego, CA, USA) to generate the Kaplan–Meier plot (log-rank $\chi^2$ test).

## Acknowledgements

This research was supported by National Key Research and Development Program of China (2022YFF1000304), National Natural Science Foundation of China (32022085/31930113/32373158), Southern Marine Science and Engineering Guangdong Laboratory (Zhuhai) (SML2023SP234), China Postdoctoral Science Foundation (2023M743987), and Guangdong Basic and Applied Basic Research Foundation (2023A1515110528). The funders had no role in study design, data collection and analysis, decision to publish, or preparation of the manuscript.

## Additional information

### Funding

| Funder | Grant reference number | Author |
| --- | --- | --- |
| National Key Research and Development Program of China | 2022YFF1000304 | Jianguo He |
| National Natural Science Foundation of China | 32022085 | Chaozheng Li |
| Southern Marine Science and Engineering Guangdong Laboratory | SML2023SP234 | Chaozheng Li |
| China Postdoctoral Science Foundation | 2023M743987 | Bang Xiao |
| Guangdong Basic and Applied Basic Research Foundation | 2023A1515110528 | Bang Xiao |
| National Natural Science Foundation of China | 31930113 | Chaozheng Li |
| National Natural Science Foundation of China | 32373158 | Chaozheng Li |

The funders had no role in study design, data collection and interpretation, or the decision to submit the work for publication.

### Author contributions

Bang Xiao, Data curation, Formal analysis, Funding acquisition, Validation, Visualization, Methodology, Writing – original draft; Shihan Chen, Formal analysis, Validation, Methodology; Yue Wang, Data curation, Validation, Methodology; Xuzheng Liao, Validation, Investigation, Methodology; Jianguo He, Funding acquisition, Investigation, Project administration; Chaozheng Li, Conceptualization, Resources, Funding acquisition, Investigation, Methodology, Project administration, Writing – review and editing

### Author ORCIDs

Chaozheng Li ⓘD https://orcid.org/0000-0001-7639-7831

### Ethics

All animal experiments were approved by Institutional Animal Care and Use Committee of Sun Yat-Sen University (Approval No. SYSU-IACUC-2023-B0005).

Reviewer #1 (Public review): https://doi.org/10.7554/eLife.101460.3.sa1
Reviewer #3 (Public review): https://doi.org/10.7554/eLife.101460.3.sa2
Author response https://doi.org/10.7554/eLife.101460.3.sa3

## Additional files

### Supplementary files

Supplementary file 1. Sequences of the primers used in this study.

Supplementary file 2. Heat shock proteins downregulated DEGs by transcriptome sequencing.

Supplementary file 3. Partial promoter of AMPs was used in this study.

MDAR checklist

### Data availability

Sequence data has been uploaded under Bioproject PRJNA1110613. Source data files have been provided for *Figures 1–8*.

The following dataset was generated:

| Author(s) | Year | Dataset title | Dataset URL | Database and Identifier |
|---|---|---|---|---|
| Xiao B | 2024 | Heat Shock Factor Regulation of Antimicrobial Peptides Expression Suggests a Conserved Defense Mechanism Induced by Febrile Temperature in Arthropods | https://www.ncbi.nlm.nih.gov/bioproject/PRJNA1110613/ | NCBI BioProject, PRJNA1110613 |

The following previously published dataset was used:

| Author(s) | Year | Dataset title | Dataset URL | Database and Identifier |
|---|---|---|---|---|
| Xiao B | 2023 | Mechanism of high temperature induced shrimp resistance to white spot syndrome virus | https://www.ncbi.nlm.nih.gov/bioproject/?term=PRJNA1050424 | NCBI BioProject, PRJNA1050424 |

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
