## [Editor Report · eLife Assessment]

This is an **important** study that addresses the role of fever as a conserved response to viral infection. It demonstrates that the heat-shock factor, HSF1, is activated by increased temperature during fever to enhance the anti-viral immune response. The data provides **compelling** evidence for the conclusions and the work will be of interest to virologists, immunologists, and cell biologists.

---

## [Referee Report · Reviewer #1 (Public review)]

Summary:

In the manuscript "Heat Shock Factor Regulation of Antimicrobial Peptides Expression Suggests a Conserved Defense Mechanism Induced by Febrile Temperature in Arthropods," Xiao and colleagues examine the role of the shrimp Litopenaeus vannamei HSF1 ortholog (LvHSF1) in the response to viral infection. The authors provide compelling support for their conclusions that the activation of LvHSF1 limits viral load at high temperatures. Specifically, the authors convincingly show that (i) LvHSF1 mRNA and protein are induced in response to viral infection at high temperatures, (ii) increased LvHSF1 levels can directly induce the expression of the nSWD (and directly or indirectly other antibacterial peptides, AMPs), (ii) nSWD's antimicrobial activities can limit viral load, and, (iv) LvHSF1 protects survival at high temperatures following virus infection. These data thus provide a model by which an increase in HSF1 levels limits viral load through the transcription of antimicrobial peptides, and provide a rationale for the febrile response as a conserved response to viral infection.

Strengths:

The large body of careful time series experiments, tissue profiling, and validation of RNA-seq data is convincing. Several experimental methodologies are used to support the author's conclusions that nSWD is an LvHSf1 target and increased LvHSF1 alone can explain increased levels of nSWD. Similar carefully conducted experiments also conclusively implicate nSWD protein in limiting WSSV viral loads.

Weaknesses:

As with any complex biological phenomenon, several aspects remain incompletely explained. Nevertheless, in their revision, the authors provide additional analyses supporting the authors model that losing LvHSF1 is not detrimental to survival, by more directly altering viral loads. In addition, their revised manuscript clarifies the complex interactions between infection, the role of HSF1, and hormesis. These revisions increase the impact of their findings.

Comments on revisions:

The authors have addressed all comments, and the manuscript is very much improved.

---

## [Referee Report · Reviewer #3 (Public review)]

In the manuscript titled "Heat Shock Factor Regulation of Antimicrobial Peptides Expression Suggests a Conserved Defense Mechanism Induced by Febrile Temperature in Arthropods", the authors investigate the role of heat shock factor 1 (HSF1) in regulating antimicrobial peptides (AMPs) in response to viral infections, particularly focusing on febrile temperatures. Using shrimp (Litopenaeus vannamei) and Drosophila S2 cells as models, this study shows that HSF1 induces the expression of AMPs, which in turn inhibit viral replication, offering insights into how febrile temperatures enhance immune responses. The study demonstrates that HSF1 binds to heat shock elements (HSE) in AMPs, suggesting a conserved antiviral defense mechanism in arthropods. The findings are informative for understanding innate immunity against viral infections, particularly in aquaculture. However the logical flow of the paper can be improved.

Comments on revisions:

Some aspects of the initial study design, regarding the selection of representative candidate genes and the logical flow, raised concerns. However, these issues have been addressed in the revised manuscript through additional validations and clarifications. Most of my comments and concerns were sufficiently addressed in the revised manuscript. The results support the authors' conclusion that HSF1-dependent regulation of AMP expression contributes to antiviral defense under febrile conditions.

---

## [Author Response]

The following is the authors’ response to the original reviews.

**Public Reviews:**

**Reviewer #1 (Public review):**
Weaknesses:Despite this compelling data regarding the protective role of HSF1 in the febrile response, what remains unexplained and complicates the authors' model is the observation that losing LvHSF1 at 'normal' temperatures of 25 ℃ is not detrimental to survival, even though viral loads increase and nSWD is likely still subject to LvHSF1 regulation. These observations suggest that WSSV infection may have other detrimental effects on the cell not reflected by viral load and that LvHSF1 may play additional roles in protecting the organism from these effects of WSSV infection, such as perhaps, perturbations to protein homeostasis. This is worth discussing, especially in light of the rather complicated roles of hormesis in protection from infection, the role of HSF1 in hormesis responses, and the findings from other groups that the authors discuss.

We are grateful for your unbiased advice by reviewer. And we have added the description about the role of HSF1 in hormesis responses in discussion in Lines 422-425 in the revised manuscript. Thank you.

**Reviewer #2 (Public review):**
Temperature is a critical factor affecting the progression of viral diseases in vertebrates and invertebrates. In the current study, the authors investigate mechanisms by which high temperatures promote anti-viral resistance in shrimp. They show that high temperatures induce HSF1 expression, which in turn upregulates AMPs. The AMPs target viral envelope proteins and inhibit viral infection/replication. The authors confirm this process in drosophila and suggest that there may be a conserved mechanism of high-temperature mediated anti-viral response in arthropods. These findings will enhance our understanding of how high temperature improves resistance to viral infection in animals.The conclusions of this paper are mostly well supported by data, but some aspects of data analysis need to be clarified and extended. Further investigation on how WSSV infection is affected by AMP would have strengthened the study.

We are grateful for your unbiased advice by reviewer. We have provided additional experimental evidence and supplementary instructions in the revised manuscript. Thank you.

**Reviewer #3 (Public review):**
In the manuscript titled "Heat Shock Factor Regulation of Antimicrobial Peptides Expression Suggests a Conserved Defense Mechanism Induced by Febrile Temperature in Arthropods", the authors investigate the role of heat shock factor 1 (HSF1) in regulating antimicrobial peptides (AMPs) in response to viral infections, particularly focusing on febrile temperatures. Using shrimp (*Litopenaeus vannamei*) and Drosophila S2 cells as models, this study shows that HSF1 induces the expression of AMPs, which in turn inhibit viral replication, offering insights into how febrile temperatures enhance immune responses. The study demonstrates that HSF1 binds to heat shock elements (HSE) in AMPs, suggesting a conserved antiviral defense mechanism in arthropods. The findings are informative for understanding innate immunity against viral infections, particularly in aquaculture. However, the logical flow of the paper can be improved.

We are grateful for the positive comments and the unbiased advice by reviewer. We have improved the logical flow of the paper and added corresponding instructions in the revised manuscript. Thank you.

**Recommendations for the authors:**

**Reviewer #1 (Recommendations for the authors):**
(1) Figure 1: The analysis compares Group TW to Group W (not the other way around).

Thank you very much. To uncover the molecular mechanisms by which high temperature restricts WSSV infection, two shrimp groups, Group TW and Group W, were cultured at 25 °C. Group W comprised shrimp injected with WSSV and maintained at 25 °C continuously. In contrast, Group TW was subjected to a temperature increase to 32 °C at 24 hours post-injection (hpi). Gill samples were collected for analysis 12 hours post-temperature rise (hptr) and subjected to Illumina sequencing (Figure 1A). RNA-seq was used to identify genes responsive to high temperature, particularly those encoding potential transcriptional regulators. Thank you.

(2) The RNA-seq data in Figure 1 focus only on the TFs. The manuscript would benefit from showing all the RNA-seq data and the differentially expressed genes. In particular, are the AMPs upregulated at the same time point? This should not be the case if LvHSF1 were responsible for the transcription of the AMPs, given the time lag between transcription and translation.

Thank you for your suggestion. In Author response image 1, our previous study has revealed that classical heat shock proteins (such as HSP21, HSP70, HSP60, HSP83, HSP90, HSP27, HSP10, and Bip) were induced by RNA-seq between Group TW and Group W, suggesting heat shock proteins exert a crucial role in enhancing the resistance of shrimp to WSSV at elevated temperatures (32 ℃) and underscoring the reliability of our transcriptomic findings (Xiao et al., 2024).

Additionally, we also analyzed the AMPs expression between Group TW and Group W, and the results show that some antimicrobial peptides such as Lysozyme and C-type lectin are upregulated between Group TW and Group W. Notably, we did not detect upregulated expression of SWD between Group TW and Group W. We agree with the reviewer's point of view that there is a time lag between transcription and translation. Supplementary experimental evidences show that the expression level of LvHSF1 is strongly induced by WSSV stimulation, and then the expression level of SWD begins to increase. We have added a description in Lines 136-138 in the revised manuscript.

**Author response image 1. sa3fig1:** The Figure of the heat shock proteins in Group TW and Group W.

**Author response image 2. sa3fig2:** Transcriptional expression levels of HSF1 and SWD after WSSV stimulation.

Reference:

Xiao, B., Wang, Y., He, J., Li, C., 2024. Febrile Temperature Acts through HSP70-Toll4 Signaling to Improve Shrimp Resistance to White Spot Syndrome Virus. J Immunol 213, 1187-1201.

(3) The data showing the tissue distribution of LvHSF1 and nSWD is a rigorous approach and adds to the manuscript. A similar approach to understanding the time course of expression of AMPs in relationship to LvHSF1 expression levels would strengthen the authors' conclusions that LvHSF1 induction in response to high temperatures and viral infection, in turn, upregulates SWD and other antibacterial genes.

Thank you for your suggestion. As you good suggestion, we detected the transcriptional expression levels of HSF1 and SWD after WSSV stimulation for 0, 2, 4, 6, 8, 12, 16, 20, and 24 hours. The transcriptional expression level of SWD was set to 1.00 at 0 h, in the early stage of WSSV infection (0-12 h, except 6 h), the expression level of LvHSF1 is strongly induced, and then the expression level of SWD begins to increase. Theses results show that LvHSF1 induction in response to viral infection, in turn, upregulates SWD and other antibacterial genes. Thank you.

(4) The data (Figures 3 and 4) show that LvHSF1 is necessary to survive WSSV infection at high temperatures but does not affect survival at lower temperatures, even though LvHSF1 limits VP28 levels, and viral load at both temperatures is confusing. Does this suggest that LvHSF1 is not primarily important for protection against the virus but instead, for protection from the heat-induced damage caused by high temperatures, which would not be surprising? The manuscript would benefit if the authors could address this point. How do the authors envision the protection conferred by LvHSF1 only at high temperatures?

Thank you for your comment. Although no significant difference in shrimp survival rates was observed between LvHSF1-silenced shrimp and GFP-silenced shrimp at low temperature (25 °C), shrimp with silenced LvHSF1 exhibited increased viral loads in hemocytes and gills, suggesting that upregulation of HSF1 expression can protect shrimp from WSSV infection.

Notably, the tolerance temperature for *L. vannamei* growth ranges from 7.5 to 42 °C. When infected with WSSV, shrimp use behavioral fever to elevate their body temperature (~32 °C), thereby inhibiting WSSV infection (Rakhshaninejad et al., 2023; Xiao et al., 2024). And this temperature (~32 °C) will not cause heat-induced damage to the shrimp. Our results demonstrate that febrile temperatures induce HSF1, which in turn upregulates antimicrobial peptides (AMPs) that target viral envelope proteins and inhibit viral replication.

Only at high temperatures, we observed that knockdown of HSF1 did not affect shrimp survival rate (Figure 4A). Thank you again for your valuable feedback.

Reference:

Rakhshaninejad, M., Zheng, L., Nauwynck, H., 2023. Shrimp (Penaeus vannamei) survive white spot syndrome virus infection by behavioral fever. Sci Rep 13, 18034.

Xiao, B., Wang, Y., He, J., Li, C., 2024. Febrile Temperature Acts through HSP70-Toll4 Signaling to Improve Shrimp Resistance to White Spot Syndrome Virus. J Immunol 213, 1187-1201.

(5) Related to the previous comment, the authors do not clearly distinguish between basal effects of LvHSF1 or nSWD induction and heat-induced effects and the differences related to the requirement of LvHSF1 for protection. Simply increasing LvHSF1 levels can result in increased nSWD. SWD levels increase upon WSSV infection even at 25 ℃, and the knockdown experiments suggest that this could also occur through LvHSF1. It would be useful to explicitly differentiate between basal functions of HSF1 and induced functions.

Thank you for your suggestion. In previous responses, we have distinguished between basal effects of LvHSF1 or nSWD induction and heat-induced effects.

As your good suggestion, we injected GST or rHSF1 protein into shrimp, the results showed that recombinant protein HSF1 could significantly induced the expression level of SWD (Supplementary Fig. 5C). Further, after knockdown of SWD, shrimp were injection with rLvHSF1 mixed with WSSV. The results showed that the viral load was significantly lower than the control group 48 hours post WSSV infection (Supplementary Fig. 5D). We have added these results to the Supplementary Figure 5C&5D and added a description in Lines 253-255 and Lines 290-293 in the revised manuscript. Thank you for your constructive comments.

**Reviewer #2 (Recommendations for the authors):**
(1) Two temperatures are used in the experiments of shrimp. It seems that HSF1 is also upregulated by WSSV infection at 25 ℃. However, this upregulation seems not to be able to protect the animals. The authors compare the infection at 25 and 32 ℃ but did not discuss the findings.

Thank you for your comment. Although no significant difference in shrimp survival rates was observed between LvHSF1-silenced shrimp and GFP-silenced shrimp at low temperature (25 °C), shrimp with silenced LvHSF1 exhibited increased viral loads in hemocytes and gills, suggesting that upregulation of HSF1 expression can protect shrimp from WSSV infection. We have added a discussion of this finding in Lines 461-464 in the revised manuscript. Thank you.

(2) In the abstract the authors say that "These insights provide new avenues for managing viral infections in aquaculture and other settings by leveraging environmental temperature control." However, this point has not been discussed in the main text.

We appreciated your comments. We have added a discussion about the environmental temperature control in Lines 512-514 in the revised manuscript. Thank you.

(3) Line 142: "These results suggest that LvHSF1 may play a key role in enhancing shrimp resistance to WSSV at elevated temperatures." Although this type of conclusion has been made in many studies, I think it is impossible to see a "KEY role" based mainly on change in expression.

Thank you for your suggestion. We have revised this conclusion in the revised manuscript. Thank you.

(4) Section 2.1 Induction of Heat Shock Factor 1 in Response to WSSV at High TemperatureFigure 1. Identification of HSF1 as a key factor induced by high temperature.The two titles are confusing. Whether the upregulation of HSF1 is a response to high temperature or WSSV infection? I think it is more likely a response to high temperature. Did the authors see the difference in HSF1 expression in shrimp with and without WSSV infection at high temperatures?

Thank you for your comment. We have modified the title of Section 2.1 in the revised manuscript. As your good suggestion, we have measured the expression of LvHSF1 after WSSV challenge at high temperatures (32 ℃) in revised Figure 2F-2H in Line 122 in the revised manuscript. The results demonstrate that the expression of LvHSF1 is strongly induced by WSSV stimulation at high temperatures (32 ℃) in the revised manuscript. Thank you.

(5) Figure 2. Upregulation of LvHSF1 in shrimp challenged by WSSV at both low and high temperatures. Results for WSSV challenge at high temperatures are not included in this figure.

Thank you for your suggestion. As your good suggestion, we have measured the expression of LvHSF1 after Poly (I: C) and WSSV challenge at high temperatures (32 ℃) in revised Figure 2C-2H. The results demonstrate that the expression of LvHSF1 is strongly induced by Poly (I: C) and WSSV stimulation at high temperatures (32 ℃). And we have added a description in Lines 168-179 in revised manuscript. Thank you.

(6) Section 2.2 Expression Profiles of LvHSF1 in Shrimp Under Varied Temperature Conditions and WSSV Challenge. Did the authors try poly IC and WSSV challenge at 32℃, and compare with the un-challenge group? Why were only low temperature was analyzed?

Thank you for your suggestion. As your good suggestion, we have measured the expression of LvHSF1 after Poly (I: C) and WSSV challenge at high temperatures (32 ℃) in revised Figure 2C-2H. And we have added a description about the expression of LvHSF1 after Poly (I: C) and WSSV challenge at high temperatures (32 ℃) in Lines 168-179 in revised manuscript. Thank you.

(7) Figure 2: Please indicate the temperature used in C-E and F-H in the figure legend. Statistical significance: compared with which group? Please provide information in the legend or show it in the bar chart.

Thank you for your suggestion. We have added the description of temperature used in revised Figures 2C-2E. The expression changes of HSF1 were compared with those of PBS control group at the corresponding time and we modified the comparison method of significance in revised Figures 2C-2E. Thank you.

(8) Figure 3H: There are two groups (dsGFP+PBS; dsHSF1+PBS) showing with the same symbol (dot line).

Thank you for your comment. The revised Figure 3H has used different symbols to distinguish the two groups. Thank you.

(9) Line 205: qPCR

Thank you for your careful checks. We have corrected this error in the revised manuscript. Thank you.

(10) Figure 5d and f: Please indicate the sample in each row.

Thank you for your suggestion. We have marked the samples in each row in the revised Figures 5d&5f.

(11) Figure 3 and Figure 4: Why different tissues were analyzed in the two experiments? Low temperature: gill and hemocytes. High temperature: gill and muscle? It is better to use the same tissues so that they can be compared. Please indicate the tissue analyzed in D and d.

Thank you for your suggestion. We have repeated the experiment to detect the copy number of WSSV in hemocyte at high temperature (32 °C) after LvHSF1 knockdown. The results showed that knockdown LvHSF1 showed increased viral loads in shrimp hemocyte (Figure 4C). We have supplemented the tissue information in Figure 4D&4d. Thank you.

(12) Figure 2A The time for temperature treatment? hours or days?

Thank you for your comment. Transcriptional expression of LvHSF1 in different tissues of healthy shrimp subjected to low (25 °C) and high (32 °C) temperatures for 12 hours. We have supplemented this information in the legend of Figure 2A in Lines 840-841 in revised manuscript. Thank you.

(13) Line 249: purified by SDS-PAGE gel?

Thank you for your comment. We have modified this description in Lines 272-274 in current manuscript. Thank you.

(14) Line 258 "Next, to verify whether the anti-WSSV function of nSWD was mediated by LvHSF1 at high temperature". I think it is confusing to use "mediated" here. It seems that HSF1 is downstream of nSWD. Actually, HSF1 controls the expression of nSWD and thus regulates the anti-WSSV effect of shrimp at high temperatures.

We appreciated your comments. We have modified this description in Lines 282-283 in current manuscript. Thank you.

(15) Line 458 "The most probable anti-WSSV mechanism of nSWD is its direct interaction with WSSV envelope proteins VP24 and VP26, potentially inhibiting viral entry into target cells. I suggest the author analyze the entry of WSSV to see whether nSWD blocks this process.

Thank you for your comment. In general, the antimicrobial mechanism of action of AMPs is thought to involve direct membrane disruption, especially for enveloped virus (such as WSSV) (Wilson et al., 2013).

Thanks to the reviewers for their valuable comments. Our manuscript mainly focuses on the febrile temperature-inducible HSF in host antiviral immunity, and the role of HSF1 in regulating antimicrobial effectors (such as SWD). Due to the limitation of the manuscript's length, we will further investigate the functional mechanisms of SWD-specific anti-WSSV in future studies. Thank you.

Reference:

Wilson, S.S., Wiens, M.E., Smith, J.G., 2013. Antiviral Mechanisms of Human Defensins. Journal of Molecular Biology 425, 4965-4980.

(16) Line 435-456 The author discusses the difference between two shrimp species. Did the two studies measure the same immune parameters? I wonder whether the different observation is due to true differences or different methods they used to evaluate the response. If no immune response was promoted in the previous study, what's the possible anti-viral mechanism?

We appreciated your comments. Firstly, the shrimps in the two experimental groups have different adaptability to temperature. The optimal water temperature for *M. japonicus* growth ranges from 25 to 32 °C, and the tolerance temperature for *L. vannamei* growth ranges from 7.5 to 42 °C. Secondly, the experimental environmental factors are different in the two experimental groups. Ammonia is a key stress factor in aquatic environments that usually increases the risk of pathogenic diseases in aquatic animals, however, High temperatures (32°C) have been shown to inhibit the replication of WSSV and reduce mortality in WSSV-infected shrimp. Thirdly, the two studies tested different immune indicators. Ammonia-induced Hsf1 suppressed the production and function of MjVago-L, an arthropod interferon analog. In this study, our findings revealed the molecular mechanism through which the HSF-AMPs axis mediates host resistance to viruses induced by febrile temperature. Taken together, the benefits of HSF1 can be attributed to either the host or the pathogen, depending on the nature and context of the host-virus-environment interaction.

(17) Line 472 "directly bind to WSSV envelope proteins and inhibit WSSV proliferation"I think it is confusing to use "proliferation" here. It seems that the binding of HSF affects the replication process. However, based on the authors' discussion, HSF may likely block viral entry.

Thank you for your suggestion. We have modified this description in Lines 505-507 in the current manuscript. Thank you.

**Reviewer #3 (Recommendations for the authors):**
In the manuscript titled "Heat Shock Factor Regulation of Antimicrobial Peptides Expression Suggests a Conserved Defense Mechanism Induced by Febrile Temperature in Arthropods", the authors investigate the role of heat shock factor 1 (HSF1) in regulating antimicrobial peptides (AMPs) in response to viral infections, particularly focusing on febrile temperatures. Using shrimp (Litopenaeus vannamei) and Drosophila S2 cells as models, this study shows that HSF1 induces the expression of AMPs, which in turn inhibit viral replication, offering insights into how febrile temperatures enhance immune responses. The study demonstrates that HSF1 binds to heat shock elements (HSE) in AMPs, suggesting a conserved antiviral defense mechanism in arthropods. The findings are informative for understanding innate immunity against viral infections, particularly in aquaculture. However, the logical flow of the paper can be improved. Following are my specific concerns.Major comments(1) The study design is pretty good, but the logical flow is not. The following should be improved.(a) In Figure 1, the reason for selecting HSF1 as the focus of the study is not clearly explained.

Thank you for your comment. In a previous study, we have revealed that heat shock proteins exerted a significant role in enhancing the resistance of shrimp to WSSV at elevated temperature (32 ℃) (Xiao et al., 2024). GO functional enrichment analysis of DEGs between group TW and group W, indicating that most DEGs were involved in biological processes such as protein refolding, chaperone-mediated protein folding, and heat response. Therefore, special attention has been paid to heat shock factor 1 (HSF1), the master regulator of the heat shock response. We have added the description in Lines 136-138 in the revised manuscript. Thank you.

Reference:

Xiao, B., Wang, Y., He, J., Li, C., 2024. Febrile Temperature Acts through HSP70-Toll4 Signaling to Improve Shrimp Resistance to White Spot Syndrome Virus. J Immunol 213, 1187-1201.

(b) As the authors draw models in Figure 9, the established activation mechanism of HSF1 is via trimerization by the release of HSP90, which binds to misfolded proteins under stress conditions, such as heat shock. Therefore, the increase in the HSF1 mRNA level in Figure 1 is strange. The authors need to clarify this issue by explaining this established activation mechanism of HSF1 and also must provide the basis of upregulation of HSF1 by mRNA increase via citing papers in the Introduction.

We appreciated your comments. Under non-stress conditions, HSF monomers are retained in the cytoplasm in a complex with HSP90. During the stress response, such as high temperature, HSF dissociates from the complex, trimerizes, and converts into a DNA-binding conformation through regulatory upstream promoter elements known as heat shock elements (HSEs) (Andrasi et al., 2021). Previous studies have demonstrated that the expression of HSF1 was remarkably induced by stress response, such as high temperature (Ren et al., 2025), virus infection (Merkling et al., 2015), and ammonia stress (Wang et al., 2024). Our results also showed that the expression of LvHSF1 was significant induced by WSSV infection and high temperature (Figure 2). Therefore, this is not surprising that the increase in the HSF1 mRNA level in Figure 1.

In response, we have revised the proposed model to better reflect our experimental findings and the accompanying description. This revision ensures that the schematic is consistent with our data and accurately represents the proposed mechanism. We appreciate your careful review and constructive feedback.

Reference:

Andrasi, N., Pettko-Szandtner, A., Szabados, L., 2021. Diversity of plant heat shock factors: regulation, interactions, and functions. J Exp Bot 72, 1558-1575.

Ren, Q., Li, L., Liu, L., Li, J., Shi, C., Sun, Y., Yao, X., Hou, Z., Xiang, S., 2025. The molecular mechanism of temperature-dependent phase separation of heat shock factor 1. Nature Chemical Biology.

Merkling, S.H., Overheul, G.J., van Mierlo, J.T., Arends, D., Gilissen, C., van Rij, R.P., 2015. The heat shock response restricts virus infection in Drosophila. Sci Rep 5, 12758.

Wang, X.X., Zhang, H., Gao, J., Wang, X.W., 2024. Ammonia stress-induced heat shock factor 1 enhances white spot syndrome virus infection by targeting the interferon-like system in shrimp. mBio 15, e0313623.

(c) For RNA seq analysis in both in Figures 1 and 5, they need to provide changes in conventional HSF1 target chaperones (many HSPs) to validate their RNA seq data.

Thank you for your suggestion. In Authopr response image 1, our previous study has revealed that classical heat shock proteins (such as HSP21, HSP70, HSP60, HSP83, HSP90, HSP27, HSP10, and Bip) were induced by RNA-seq between Group TW and Group W, suggesting heat shock proteins exert a crucial role in enhancing the resistance of shrimp to WSSV at elevated temperatures (32 ℃) and underscoring the reliability of our transcriptomic findings (Xiao et al., 2024). We have added the description in Lines 136-138 in the revised manuscript.

In Figure 5, we have supplemented the heat shock proteins downregulated DEGs by transcriptome sequencing of dsGFP +WSSV (32 ℃) vs. dsLvHSF1 +WSSV (32 ℃) in Supplementary table 2. The results showed that the classical heat shock proteins were downregulated by the RNA-seq, underscoring the reliability of our transcriptomic findings. We have added the description in Lines 213-216 in the revised manuscript. Thank you.

Reference:

Xiao, B., Wang, Y., He, J., Li, C., 2024. Febrile Temperature Acts through HSP70-Toll4 Signaling to Improve Shrimp Resistance to White Spot Syndrome Virus. J Immunol 213, 1187-1201.

(d) In Figure 5, they did experiments by focusing on the changes by HSF1 knockdown at 32 ℃. However, the logical flow should be focusing on genes whose expression was increased by 32 ℃ compared with 25 ℃ (in figure 1), among them they need to characterize HSF1 target genes. Here as mentioned above, classical HSP genes must be included in addition to those AMP genes.

Thank you for your suggestion. As your good suggestion, we have supplemented the heat shock proteins downregulated DEGs by transcriptome sequencing of dsGFP +WSSV (32 ℃) vs. dsLvHSF1 +WSSV (32 ℃) in Supplementary table 2. The results showed that the classical heat shock proteins were downregulated by the RNA-seq, underscoring the reliability of our transcriptomic findings. We have added the description in Lines 213-216 in the revised manuscript. Thank you.

(e) What is the logical basis of just picking nSWD? It is another example of cherry-picking similar to picking HSF1 in Figure 1.

We appreciated your comments. To determine how temperature-induced LvHSF1 restricts WSSV infection, RNA-seq was performed to identify target genes regulated by HSF1. By analyzing the differentially expressed genes (DEGs), we screened eight candidate proteins for immunity-effector molecules, including SWD, CrustinⅠ, C-type lectin, Anti-lipopolysaccharide factor (ALF), and Vago. CrustinⅠ has been shown to play an important role in antiviral immunity (Li et al., 2020); C-type lectin (CTL1) can bind to the VP28, VP26, VP24, VP19, and VP14, thereby inhibiting the infection of WSSV (Zhao et al., 2009); Anti-lipopolysaccharide factor (ALF3) performs its anti-WSSV activity by binding to the envelope protein WSSV189 (Methatham et al., 2017); Vago can inhibit WSSV infection by activating the Jak/Stat pathway in shrimp (Gao et al., 2021). However, the detailed regulatory mechanism of SWD against WSSV was unclear, and particular attention was paid to the SWD. We have added the description in Lines 215-220 in the revised manuscript. Thank you for your valuable comments and the logic of the manuscript has been improved.

Reference:

Li, S., Lv, X., Yu, Y., Zhang, X., Li, F., 2020. Molecular and Functional Diversity of Crustin-Like Genes in the Shrimp Litopenaeus vannamei, Marine Drugs 18, 361.

Zhao, Z.Y., Yin, Z.X., Xu, X.P., Weng, S.P., Rao, X.Y., Dai, Z.X., Luo, Y.W., Yang, G., Li, Z.S., Guan, H.J., Li, S.D., Chan, S.M., Yu, X.Q., He, J.G., 2009. A novel C-type lectin from the shrimp Litopenaeus vannamei possesses anti-white spot syndrome virus activity. Journal of Virology 83, 347-356.

Methatham, T., Boonchuen, P., Jaree, P., Tassanakajon, A., Somboonwiwat, K., 2017. Antiviral action of the antimicrobial peptide ALFPm3 from Penaeus monodon against white spot syndrome virus. Dev Comp Immunol 69, 23-32.

Gao, J., Zhao, B.R., Zhang, H., You, Y.L., Li, F., Wang, X.W., 2021. Interferon functional analog activates antiviral Jak/Stat signaling through integrin in an arthropod. Cell Rep 36, 109761.

(f) Likewise, choosing Atta in S2 cells needs logic.

We appreciated your comments. Our manuscript revealed that febrile temperature inducible HSF1 confers virus resistance by regulating the expression of antimicrobial peptides (AMPs) in *L. vannamei*. Further, we want to know that whether HSF1 regulation of antimicrobial peptides is a conserved defense mechanism induced by elevated temperature in arthropods, and experiments were performed in an invertebrate model system (*Drosophila* S2 cells). Previous study showed that DmAMPs (such as Attacin A, Cecropins A, Defensin, Metchnikowin, and Drosomycin) exerted a significant role in the antiviral immunity in *Drosophila* (Zhu et al., 2013). Our results showed that the expression of Attacin A, Cecropins A and Defensin were remarkably induced by DmHSF, and the expression of Attacin A was the highest induced. Therefore, DmAtta was chosen as a representative to further demonstrate that DmHSF1 exerts its anti-DCV function by regulating DmAMPs. We have added the description in Lines 328-330 and Lines 361-364 in the revised manuscript. Thank you for your valuable comments and the logic of the manuscript has been improved.

Reference:

Zhu, F., Ding, H., Zhu, B., 2013. Transcriptional profiling of Drosophila S2 cells in early response to Drosophila C virus. Virol J 10, 210.

(2) From Figure 6I to 6K, the authors aimed to verify whether the anti-WSSV function of nSWD was mediated by LvHSF1 at high temperatures. However, what they showed was just showing that nSWD plays anti-WSSV function downstream of HSF1. The authors should show additional data for dsControl+rnSWD.

Thank you for your suggestion. As your suggestion, after knockdown of SWD, shrimp were injection with rLvHSF1 mixed with WSSV. The results showed that the viral load was significantly lower than the control group 48 hours post WSSV infection (Supplementary Fig. 5D). We have added these results to the Supplementary Figure 5C&5D and added a description in Lines 290-293 in the revised manuscript. Thank you for your constructive comments.

(3) For the physical interaction between nSWD and WSSV, it will be great if the authors perform Alphafold3 prediction analysis (Abramson et al PMID: 38718835).

Thank you for your suggestion. As you suggestion, we performed Alphafold3 prediction analysis on SWD and WSSV (VP24 and VP26). The predicted template modeling (pTM) score measures the accuracy of the entire structure. A pTM score above 0.5 means the overall predicted fold for the complex might be similar to the true structure. The Alphafold3 prediction results show that there is a possible interaction between SWD and WSSV. Notably, our manuscript demonstrated that rSWD could interact with VP24 and VP26 by pulldown assays and confocal analysis.

**Author response image 3. sa3fig3:** Alphafold3 prediction analysis of SWD&VP24 as follow (pTM = 0.64).

**Author response image 4. sa3fig4:** Alphafold3 prediction analysis of SWD&VP26 as follow (pTM = 0.53).

Minor comments(1) In the Abstract and many other places, the authors need to specifically write "*Drosophila* S2 cells" instead of "*Drosophila*" because conventionally Drosophila implies fruit fly as an organism. We don't say cultured human cells as "human" or "*Homo sapiens*" in papers.

Thank you for your suggestion. We have modified the description of *Drosophila* in the revised manuscript. Thank you.

(2) Figure numbers can be reduced for better readability. I would combine Figures 1 and 2, and Figures 3 and 4. If the combined figures are too crowded, some can go to into supplementary figures.

Thank you for your suggestion. We have moved the Poly (I: C) data to Supplementary Figure 2 in the revised manuscript. However, we have added some experimental data to Figures 1, 2, 3, and 4. Therefore, we did not combine Figure 1 and Figure 2, and Figures 3 and 4. Thank you.

(3) One of the best-understood roles of HSF1 in physiology other than heat shock response is longevity, in particular with *C. elegans*. The authors need to mention this in the Discussion by citing the following recent review paper (Lee PMID: 36380728).

Thank you for your suggestion. We have supplemented the description of HSF1 regulating longevity and aging of organisms and cited the above reference in the revised manuscript (Lee and Lee, 2022). Thank you.

Reference:

Lee, H., Lee, S.V., 2022. Recent Progress in Regulation of Aging by Insulin/IGF-1 Signaling in *Caenorhabditis elegans*. Mol Cells 45, 763-770.

(4) Please make your own label for small letter panels or transfer small letter panels to supplementary figures.

Thank you for your suggestion. We have adjusted the relevant letter labels. The uppercase letters represent the main image of the Figure, and the small letter panels are the corresponding supplementary instructions in the revised manuscript. Thank you.

(5) In the introduction part, I recommend changing the references for HSFs and HSR with recent ones.

Thank you for your suggestion. We have added the latest references for HSFs and HSR in the Introduction part of the revised manuscript. Thank you.

(6) In Figure 1, it is not intuitive to understand the name groups W and TW.

We appreciated your comments. We have added the description of Group W and Group TW in revised Figure 1. Group W comprised shrimp injected with WSSV and maintained at 25 °C continuously. In contrast, Group TW was subjected to a temperature increase to 32 °C at 24 hours post-injection (hpi). Gill samples were collected for analysis 12 hours post-temperature rise (hptr) and subjected to Illumina sequencing. Thank you.

(7) Please add some kinds of sequence comparisons of SWD and nSWD for readers to understand the homology.

We appreciated your comments. We have added the multiple sequence alignment of SWD proteins in shrimp species in revised Supplementary Figure 3. Highly conserved amino acid residues and cysteine and residues are highlighted in red, indicating that LvSWD is a conserved antimicrobial peptide of the Crustin family. Thank you.

(8) Naming nSWD with "newly identified" is strange as it will not be new anymore as time goes by. Please change the name.

Thank you for your suggestion. We have modified the name of nSWD to SWD in the revised manuscript. Thank you.

(9) Please write the full name for Lv (Litopenaeus vannamei), Dm (*Drosophila melanogaster*), ds (double-stranded) before using LvHSF1, DmHSF1, and dsLvHSF1.

Thank you for your comments. We have added the full name of LvHSF1, DmHSF1, and dsLvHSF1 in the revised manuscript. Thank you.

(10) In Figure 2, it will be better to transfer poly I:C data to supplementary figures.

Thank you for your comments. We have moved the Poly (I: C) data to Supplementary Figure 2 in the revised manuscript. Thank you.

(11) The label for pGL3-nSWD-M12 is confusing. M1 and M2 are OK. Please change M12 with M1/2 or another one.

Thank you for your suggestion. We have changed pGL3-nSWD-M12 with pGL3-nSWD-M1/2 in the revised manuscript. Thank you.